# Characterization of a *Toxoplasma* effector uncovers an alternative GSK3/β-catenin-regulatory pathway of inflammation

Huan He[1], Marie-Pierre Brenier-Pinchart[1], Laurence Braun[1], Alexandra Kraut[2], Bastien Touquet[3], Yohann Couté[2], Isabelle Tardieux[3], Mohamed-Ali Hakimi[1]*, Alexandre Bougdour[1]*

[1]Team Host-pathogen interactions & immunity to infection, University of Grenoble Alpes, Inserm, CNRS, IAB, Grenoble, France; [2]University of Grenoble Alpes, CEA, Inserm, BIG-BGE, Grenoble, France; [3]Team Membrane and Cell Dynamics of Host Parasite Interactions, University of Grenoble Alpes, Inserm, CNRS, IAB, Grenoble, France

**Abstract** The intracellular parasite *Toxoplasma gondii*, hijacks evolutionarily conserved host processes by delivering effector proteins into the host cell that shift gene expression in a timely fashion. We identified a parasite dense granule protein as GRA18 that once released in the host cell cytoplasm forms versatile complexes with regulatory elements of the β-catenin destruction complex. By interacting with GSK3/PP2A-B56, GRA18 drives β-catenin up-regulation and the downstream effects on host cell gene expression. In the context of macrophages infection, GRA18 induces the expression of a specific set of genes commonly associated with an anti-inflammatory response that includes those encoding chemokines CCL17 and CCL22. Overall, this study adds another original strategy by which *T. gondii* tachyzoites reshuffle the host cell interactome through a GSK3/β-catenin axis to selectively reprogram immune gene expression.
DOI: https://doi.org/10.7554/eLife.39887.001

*For correspondence:
mohamed-ali.hakimi@inserm.fr
(M-AH);
alexandre.bougdour@univ-grenoble-alpes.fr (AB)

Competing interests: The authors declare that no competing interests exist.

## Introduction

*Toxoplasma gondii* is the causative agent of toxoplasmosis, a widespread parasitic disease in humans which has been recognized as leading cause of deaths attributed to foodborne illness in the United States (*Scallan et al., 2015*). Severe to life-threatening Toxoplasmosis mainly occur in immunocompromised people, with acquired immunodeficiency syndrome or under chemo- and graft rejection therapies (*Montoya and Liesenfeld, 2004*). In addition, outcomes of congenital toxoplasmosis significantly vary with the timing of infection from recurrent eye diseases to adverse motor or neurologic impairments that can cause stillbirth (*Halonen and Weiss, 2013*). *T. gondii* belongs to the protozoan phylum Apicomplexa and as most Apicomplexa species, develops and proliferates inside a surrogate host cell. Remarkably, *T. gondii's* host range is exceptionally broad since it can infect virtually all nucleated cells of mammals, marsupials and birds (*Dubey, 2009*).

To achieve intracellular lifestyle, the invasive tachyzoite stage of *T. gondii* triggers the formation of a unique membrane-bound compartment called the Parasitophorous Vacuole (PV). The PV is shaped as a niche kept hidden from harmful endocytic processing thereby enabling tachyzoite growth and multiplication (*Jones et al., 1972*; *Mordue et al., 1999*). In the last decade, several studies have highlighted the contribution of parasite effectors delivered in the host cell at the very onset of-or post-invasion to promote folding and maturation of a functional PV (*Delorme-Walker et al., 2012*; *Hakimi et al., 2017*). Effectors are released from two specialized sets of secretory organelles including the pear-shaped rhoptries that contain the ROP16 and ROP38 effectors

and the spherical Dense Granules (DG) in which GRA15, GRA16, GRA24, TgIST, GRA6, and GRA25 proteins are stored (*Hakimi et al., 2017*). Once delivered in the host cell, these effectors either remain exposed at the cytoplasmic side of the PV Membrane (PVM) or cross the PVM and travel in the cytoplasm. Interestingly, members of the second class (i.e. GRA16, GRA24, and TgIST) have all been assigned the host nucleus as final destination where they target distinct host regulators to modulate the expression of specific sets of genes.

The comprehensive analysis of how the cohort of effectors interplay during infection at cellular and host levels remains a major challenge which implies identifying and functionally characterizing the *T. gondii* effector repertoire in given cellular and host contexts. With this concern, we searched for new effectors and characterized GRA18 as the first DG protein that strictly remains in the host cell cytoplasm once delivered from the PV-enclosed tachyzoite. We provide evidence that the exported GRA18 is part of multi-partner - that is, versatile - complexes, specifically formed with components of the β-catenin destruction complex, which includes β-catenin, GSK3α/β, and the PR56/B'-containing PP2A holoenzyme and as such prevents the continual elimination of β-catenin. Accordingly, in presence of GRA18, cytoplasmic β-catenin travels to and accumulates in the host cell nucleus where it activates otherwise repressed target genes. Nuclear β-catenin is known as the main effector of the canonical Wnt signaling pathway, acting as a coactivator of the Lymphoid Enhancer-binding Factor (LEF) or T Cell Factor (TCF) proteins to drive Wnt-specific transcriptional programs depending on cell lineages (*Cadigan and Waterman, 2012*; *Schuijers et al., 2014*). In murine macrophages, we showed that GRA18 induces in a β-catenin-dependent fashion the expression of a particular set of chemokines, that is, *Ccl17*, *Ccl22*, and *Ccl24*, notoriously known associated with anti-inflammatory effects (*Biswas and Mantovani, 2010*; *Mantovani et al., 2004*), but which have yet not been identified as β-catenin targets. In addition to discovering *T. gondii* GRA18, its partners and the down signaling pathway in the course of infection, this work also uncovers an unexpected involvement of β-catenin during the resolution of inflammatory processes.

## Results

### GRA18 is secreted and exported to the cytoplasm of infected host cells

The gene *TGGT1_288840,* hereafter referred as *GRA18,* was originally found along with the previously characterized genes *GRA16, GRA24, and TgIST* in an in silico search for candidate genes encoding proteins delivered by tachyzoites into host cells (*Bougdour et al., 2013*; *Braun et al., 2013*; *Gay et al., 2016*). GRA18 protein accommodates both a signal peptide for targeting to the secretory pathway and a canonical TEXEL motif found on PV residing proteins (i.e. GRAs proteins), some of which traffic across the PVM to reach the host cell cytoplasm (*Coffey et al., 2015*; *Hakimi et al., 2017*; *Hammoudi et al., 2015*; *Hsiao et al., 2013*) (*Figure 1A*). To determine the localization of GRA18, a three hemagglutinin (HA[3]) epitope tag or HAFlag (HF)-epitope tags were inserted either at the carboxyl-terminus of the endogenous *GRA18* locus in the type I strain (RH *ku80*) or as an extra copy in type II strain (Pru *ku80*), respectively. In extracellular tachyzoites, GRA18-HF partially co-localized with the DG resident protein GRA1 but not with the micronemal protein MIC2 or the toxofilin rhoptry protein (*Figure 1B*). When expressed in intracellular replicating tachyzoites, the GRA18-HA[3] protein accumulated in the cytoplasm of infected cells over time (*Figure 1C*, upper panel), but was not detected in host cell nucleus in contrast to the yet identified effectors (i.e. PP2C-hn, ROP16, ROP47, GRA16, GRA24, GRA28, and TgIST; for review, see *Hakimi et al., 2017*). In cells infected by type II tachyzoites (Pru *ku80*) expressing the chimeric GRA18-HF under the control of the promoter of *GRA1,* a similar GRA18-HF cytoplasmic distribution was observed (*Figure 1C*, lower panel) however the chimeric construct significantly accumulated in the parasite cytoplasm and the PV space, suggesting that secretion and export are limiting steps for GRA18 trafficking. Moreover, in the absence of the parasite MYR1 protein, the export of GRA18 through the PVM no longer occurred (*Figure 1D*), indicating that GRA18 uses an export pathway shared with the other GRA effectors (*Franco et al., 2016*). GRA18 is a 70 kDa protein predicted to be partially disordered (*Figure 1E*) with no apparent homolog counterpart outside of the Coccidia. While GRA18 shows little polymorphism among the three major strain types of *T. gondii* (99% identity, *Figure 1A*), the protein is highly divergent in *Neospora caninum* (39% identity). Collectively, these data indicate that GRA18 is an additional member of the class of *T. gondii* DG proteins that

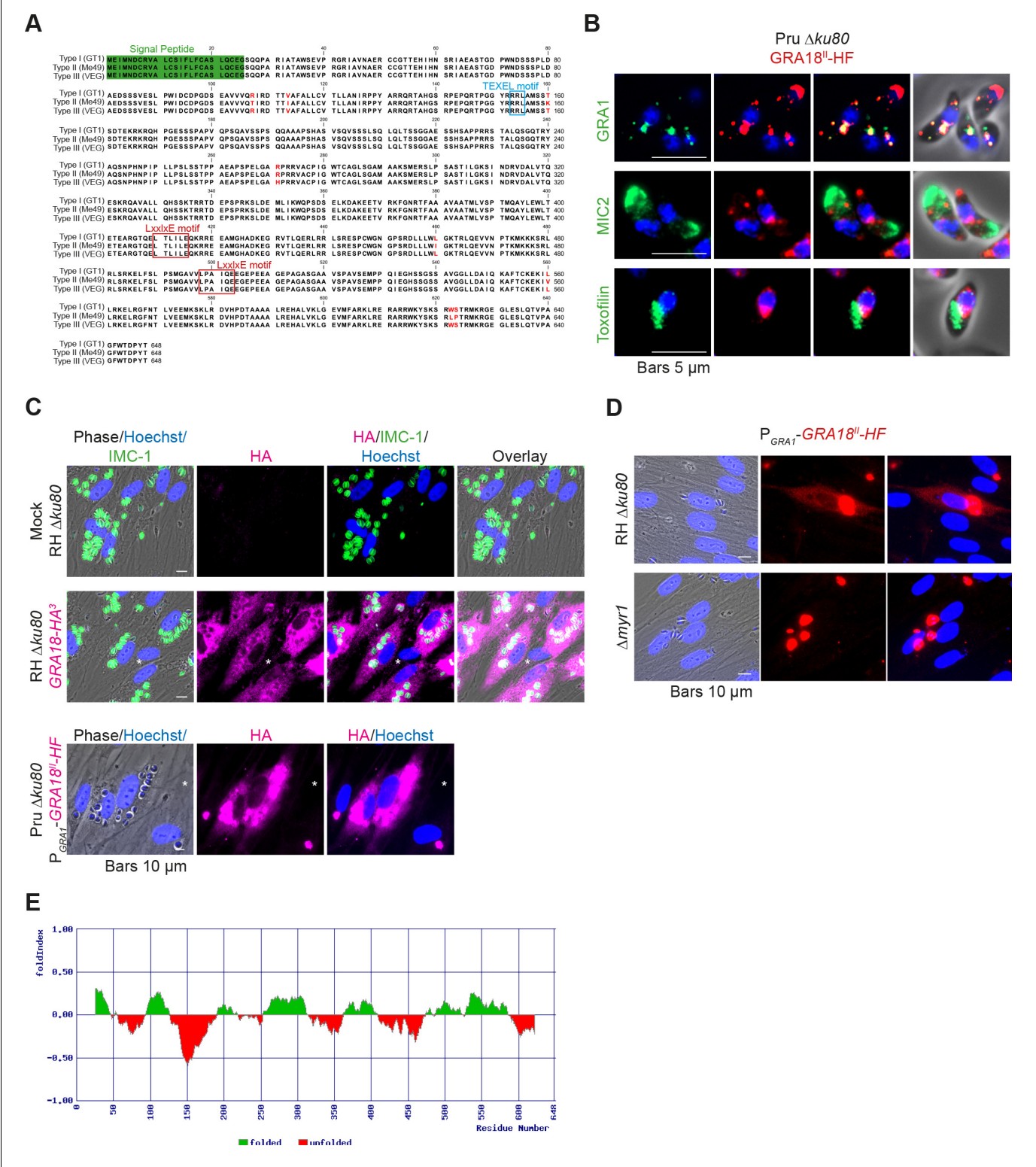

**Figure 1.** GRA18 is secreted and exported to the host cell cytoplasm. (**A**) Alignment of GRA18 alleles from *T. gondii* strains of types I (*TGGT1_288840*), II (*TGME49_288840*), and III (*TGVEG_288840*). The signal peptide sequence (highlighted in green), the B56 SLiM motifs (LxxLx; boxed in red), and the *T. gondii* Export Element (TEXEL; RRL motif, boxed in blue) are shown. Single amino acid polymorphisms are indicated by red letters. The alignment was done using ClustalW method. (**B**) GRA18[II]-HAFlag in Pru *ku80* extracellular parasites is contained in cytoplasmic organelles distinct from the apical micronemes (MIC2) and rhoptries (Toxofilin), and partially co-localizing with the dense granule protein GRA1. (**C**) GRA18 secretion and export to the

*Figure 1 continued on next page*

*Figure 1 continued*

host cytoplasm. HFFs were infected with type I RH parasites expressing endogenously tagged GRA18 with hemagglutinin (HA) (upper panel, RHΔ*ku80* GRA18-HA3, in red, Triton X-100 permeabilization) or type II Pru strain ectopically expressing a HAFlag (HF)-tagged copy of GRA18$^{II}$ under the control of the strong promoter of GRA1 (Pru Δ*ku80* P$_{GRA1}$-GRA18$^{II}$-HF, ethanol permeabilization). Cells were fixed 18 hr post-infection (hpi) and stained with anti-HA antibodies and Hoechst DNA-specific dye (in blue). The white asterisks indicate uninfected HFF cells. (D) MYR1 is required for GRA18 export in the host cell. HFFs were infected with RH WT or RH Δ*myr1* parasites transiently transfected with a vector expressing an HF-tagged GRA18 (P$_{GRA1}$-GRA18$^{II}$-HF), and at 18 hpi, the cultures were fixed and stained with antibodies to the HA tag. (E) Schematic representation of GRA18 probability of disorder. Segments with values < 0 are predicted to be disordered (in red), and segments with values > 0 correspond to folded regions (in green).
DOI: https://doi.org/10.7554/eLife.39887.002

are exported in the host cell during tachyzoite life cycle (*Hakimi et al., 2017*; *Nadipuram et al., 2016*).

## GRA18 forms complexes with host components of the β-catenin destruction complex

Since GRA18 does not carry any recognizable structural domain that can be used to infer its function, we sought for host molecular partners that could identify the host cell pathway GRA18 could interfere with. Initially, we used a genome-wide yeast-two hybrid (Y2H) screen and screened a human placenta complementary DNA (cDNA) library (referred as prey) using N-terminally fused GRA18 (aa 27 to 648) from type II strain as a bait (LexA-GRA18$^{II}$). Of a total of 65.8 million cDNA fragments screened, 65 positive hits covering 11 different proteins were found. The binding proteins were given Global Predicted Biological Score (Global PBS) ranging from A to D ('A' having the highest confidence of binding) (*Supplementary file 2*), if their coding sequences are in-frame and have no in-frame stop codons. Interestingly, multiple components of the β-catenin destruction complex were found among the interactants: β-catenin itself, the glycogen synthase kinase-3 (GSK3α/β), and the protein phosphatase 2A (PP2A) regulatory subunit B56α/β/γ/δ (*Figure 2A* and *Figure 2—figure supplement 1*). These interactions were subsequently validated by chromatography and mass spectrometry–based proteomics analysis on proteins extracted from human cells infected by the Pru P$_{GRA1}$-GRA18$^{II}$-HF *T. gondii* strain. Following Flag affinity chromatography, GRA18 co-eluted with β-catenin, GSK3α/β, and the PP2A-B56δ/ε within a quite stable complex that resisted to stringent salt and detergent conditions (0.5 M KCl and 0.1% NP-40) (*Figure 2B*). Interestingly, the identification of the scaffolding subunit PP2A65 RA and the catalytic subunit PP2Ac along with the PP2A-B56, argues for a functional PP2A holoenzyme as part of the complex. The B56 regulatory subunit presumably mediates the recruitment of the PP2A holoenzyme as it was the only PP2A subunit found by Y2H to interact directly with GRA18 (*Figure 2A* and *Supplementary file 2*). Importantly, when purified from HFFs infected by RH parasites in which *GRA18* endogenous locus was fused to the HA-Flag tags (*GRA18-HF*), GRA18 was found in a similar complex, with the exception of β-catenin that was presumably below the detection limit, (*Figure 2C*). Taken together, these data confirm GRA18 as a partner of GSK3 and the PP2A-B56 subunit and validate the relevance of these interactions in the context of infection by wild-type parasites.

We then detailed the interactions between GRA18 and the associated proteins using an inducible Flag-tagged GRA18–expressing HEK293 human cell line (T-Rex-293, *Figure 2D*). GRA18 Flag affinity coupled to Size-Exclusion Chromatography (SEC) and immunoblot analysis revealed that GRA18 forms distinct complexe(s) with β-catenin, GSK3β, and the PP2A-B56 holoenzyme ranging from 400 to over 700 kDa globular sizes (*Figure 2E*). While GSK3β and the PP2A-B56 holoenzyme eluted as discrete and overlapping complexes, in contrast, GRA18 and β-catenin both showed broader elution profiles, possibly reflecting the presence of multiple sub-complexes containing GRA18. Overall, these data confirm the multivalent partnership of GRA18 with the host proteins β-catenin, GSK3α/β, or the PP2A-B56 subunit.

## β-Catenin, GSK3α/β, and PP2A-B56 bind different protein domains of GRA18

To further detail the mode of interaction between GRA18 and the aforementioned partners, we performed a domain mapping analysis using the Y2H system and challenged the binding of GRA18 fragments to β-catenin, GSK3α/β, or the PP2A-B56 regulatory subunit in yeast (*Figure 3A*). Interestingly,

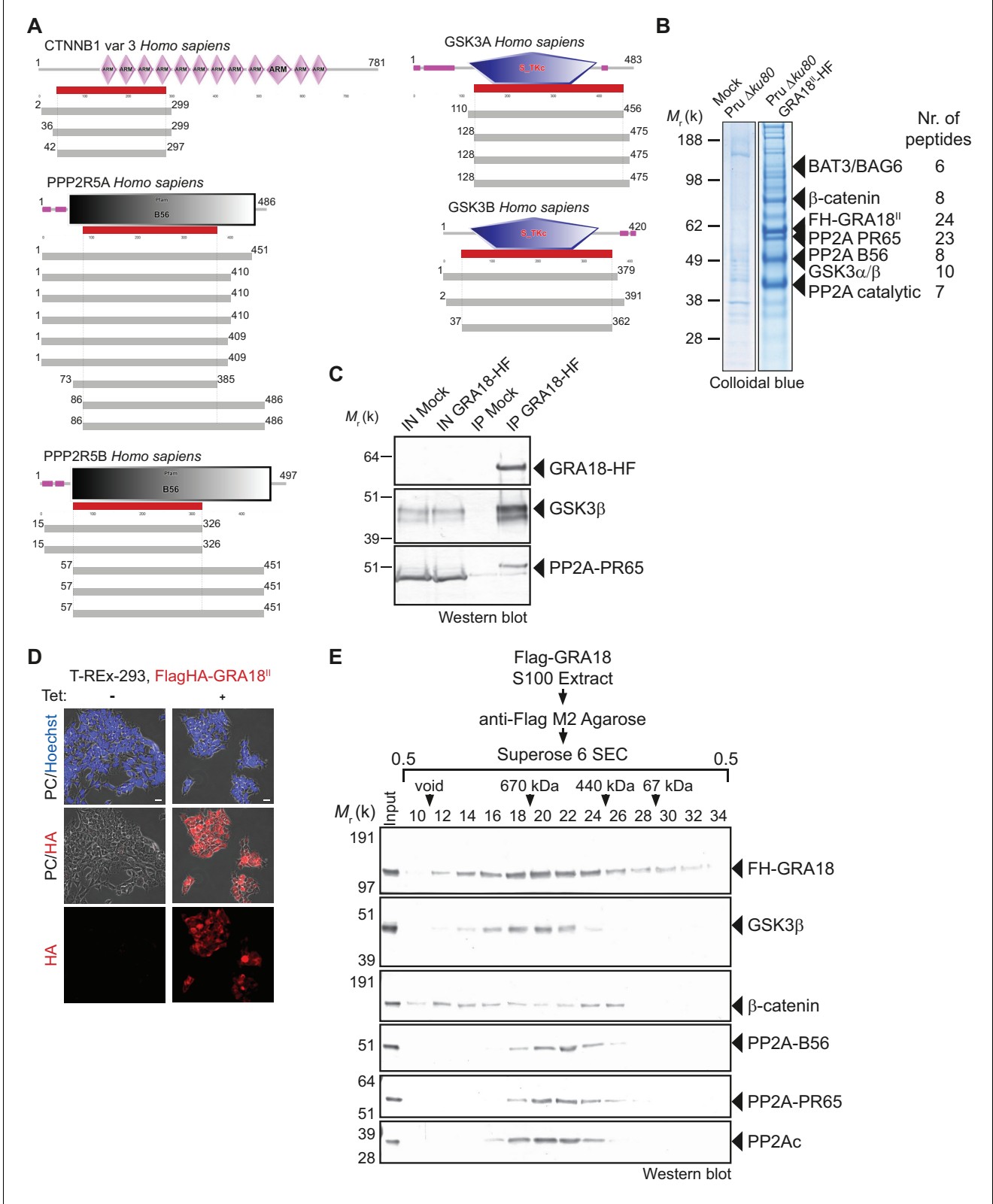

**Figure 2.** GRA18 binds directly to the host cell proteins β-catenin, GSK3α/β, and PP2A-B56. (**A**) Yeast-two hybrid screening of a human placental cDNA library to identify potential host partners for GRA18. Schematic representation of the identified partners having the highest Global PBS scores are shown. Summary of the prey clones that interacted with the GRA18 bait are represented as grey bars. Multiple independent interacting prey clones allowed Selected Interaction Domain (SID, in red) analysis that delineates the shortest fragment that is shared with all the interacting clones, and thus

*Figure 2 continued on next page*

*Figure 2 continued*

represents a potential region mediating the interaction with GRA18. (**B**) GRA18 associates with β-catenin, GSK3α/β, and PP2A-B56 in infected host cells. GRA18-associated proteins were purified by Flag affinity chromatography from protein extracts of HFF cells infected with parasites expressing HF-tagged GRA18 (Pru *k80*, P$_{GRA1}$-GRA18$^{II}$-HF). HFFs infected with Pru *ku80* parasites were used as a mock. Immunopurified proteins were resolved by SDS-PAGE, followed by colloidal blue staining and mass spectrometry analysis. The identity of the proteins and their respective number of peptides are indicated on the right of the figure. (**C**) GRA18 associates with GSK3 and PP2A-B56 when delivered to HFFs by type I RH parasites expressing endogenously tagged GRA18 with HA-Flag (RH *GRA18-HF*). IN, input; IP, immunoprecipitation. (**D**) Immunofluorescence assay (IFA) of FH-GRA18 ectopically and stably expressed in T-Rex-293 cell line. Cells were either left untreated (-) or treated with 1 µg/mL tetracycline for 12 hr before fixation and staining with anti-HA antibodies (in red) and Hoechst DNA-specific dye (in blue). Scale bar, 10 µm. (**E**) Size-Exclusion Chromatography (SEC) analysis of the GRA18-associated proteins. FH-GRA18 was immunopurified from tetracycline-induced T-Rex cells (T-Rex-GRA18FL). SEC fractions were analyzed by immunoblot using the indicated antibodies.

DOI: https://doi.org/10.7554/eLife.39887.003

The following figure supplement is available for figure 2:

**Figure supplement 1.** Summary of the hits in the Y2H screen against GRA18.

DOI: https://doi.org/10.7554/eLife.39887.004

the N-terminal region of GRA18 sheltered the binding site for β-catenin, whereas the C-terminal fragments accommodated the interaction with the PP2A-B56 subunit (*Figure 3B and D*). None of the GRA18 fragments tested other than the full-length protein could support the interactions with GSK3 in this assay, suggesting that binding to GSK3 necessitates a larger contact surface on GRA18 than the other partners.

In order to test the GRA18 binding features in a cellular context, we generated inducible T-Rex cell lines expressing each different flag tagged sub-domains of GRA18, performed immuno-tag affinity purification for each chimeric proteins and analyzed the eluates by Western Blot (*Figure 3B–C*). *Figure 3C* shows that (i) the full-length GRA18 (FH-GRA18$^{FL}$) pulled-down β-catenin, GSK3β, and PP2A-B56, (ii) the GRA18$^{Ct}$ truncated version lost the ability to bind to PP2A-B56 in agreement with the Y2H experiments, while retaining its ability to interact with GSK3β. Conversely, GRA18$^{Nt}$ induced in T-Rex cells did not interact with PP2A-B56 and its binding property to GSK3β was significantly impaired when compared to the full-length or the C-terminal region of GRA18. Collectively, these results suggest that the GRA18 C-terminal region mediates the interaction with both the phosphatase and the kinase. It is noteworthy that both the N- and C-ter truncated fragments were unable to pull down β-catenin, which was unexpected based on the Y2H results. We cannot exclude however the possibility that the harsh washing conditions (0.5 M KCl and 0.1% NP-40) shattered the weak interactions between β-catenin and the GRA18 N-terminal fragment (*Figure 3A*). Therefore, we propose that the N-terminal region of GRA18 mediates labile interaction with β-catenin that can be captured by Y2H, whereas the C-terminal region contributes to a rather strong binding to GSK3β and the PP2A-B56 holoenzyme (*Figure 3D*).

## GRA18 functions as a positive regulator of β-catenin

As demonstrated above, GRA18 interacts with well-known components of the β-catenin destruction complex, a multiprotein complex with a pivotal role in the Wnt signaling. Indeed, β-catenin regulates the transcription of the Wnt target genes. Central to the Wnt pathway is the regulation of β-catenin levels by a cytoplasmic destruction complex in which β-catenin is embedded. This complex is composed of a core scaffold protein, named the axis inhibition protein (Axin) that interacts with factors such as the adenomatous polyposis coli protein (APC), the Ser/Thr kinases GSK3, the casein kinase 1 (CK1), and the PP2A-B56 phosphatase (Reviewed by Stamos and Weis, 2013). In the absence of Wnt signaling, the destruction complex efficiently captures cytoplasmic β-catenin, leading to its phosphorylation by GSK3 and recognition by the β-TrCP ubiquitin ligase for degradation by the 26S proteasome. Wnt ligands trigger functional inactivation of the destruction complex, with ensuing escape of β-catenin from degradation, resulting in β-catenin accumulation and nuclear internalization (*Clevers, 2006*; *Li et al., 2012*; *Taelman et al., 2010*). Given the interactions between GRA18 and β-catenin, GSK3β, and PP2A-B56, we hypothesized that GRA18 could interfere with β-catenin regulation in the course of infection. To test this hypothesis, we generated parasites knockout for *GRA18* (*Figure 4A and B*) in different strain types (Pru *ku80* Δ*gra18* and 76K Δ*gra18*) which did not show any obvious growth phenotype in cell culture (*Figure 4C* and data not shown), and a somewhat

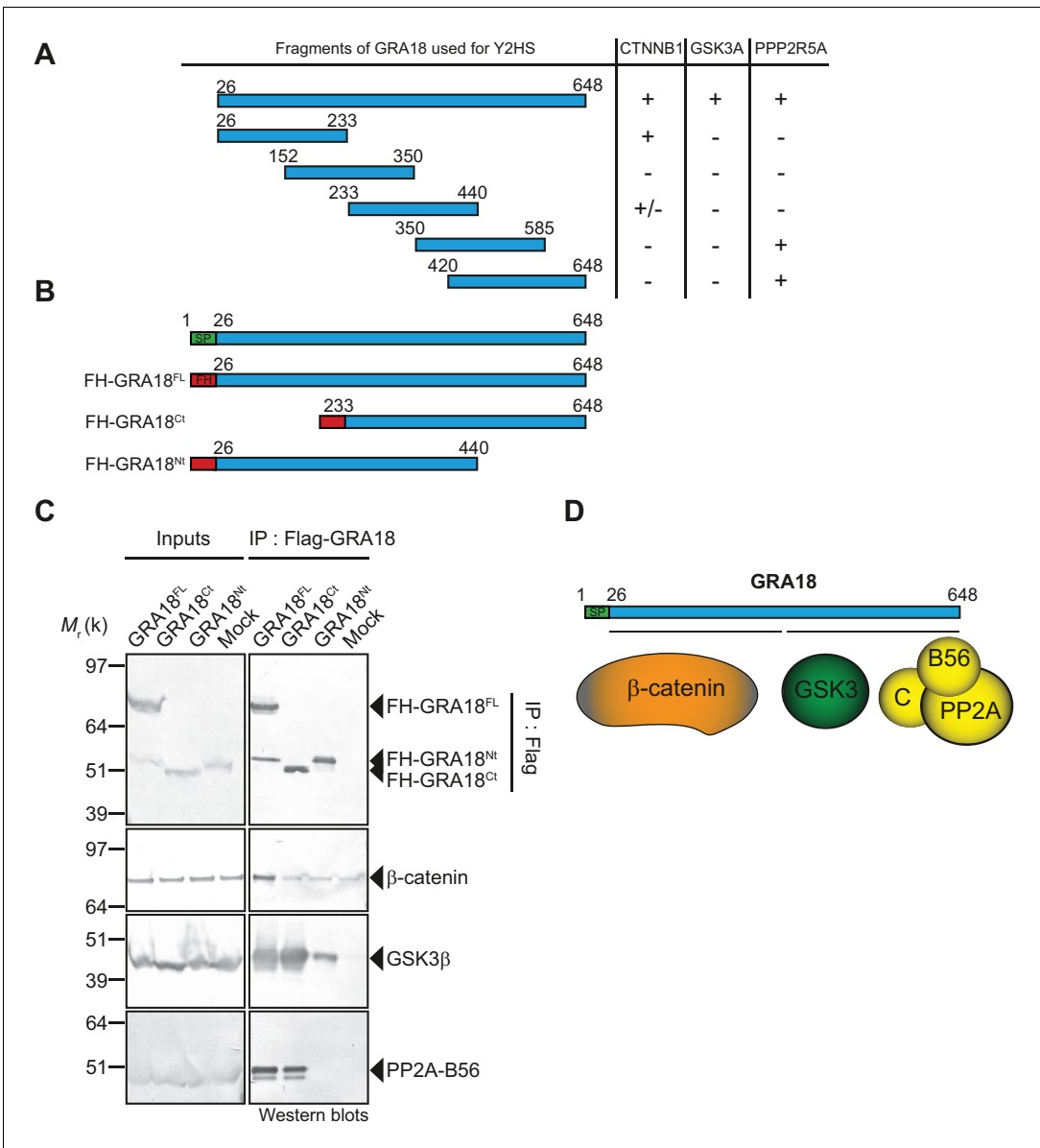

**Figure 3.** β-Catenin, GSK3, and PP2A-B56 recognize distinct domains of GRA18. (A) Interaction domain mapping by Y2H assay using the indicated fragments of GRA18 to delineate the interacting domains of GRA18 with CTNBB1 (β-catenin), GSK3A (GSK3α) and PPP2R5A (PP2A-B56α). CTNNB1 interacts with the N-terminus of GRA18, whereas PPP2R5A interacts with the C-terminal fragments. GSK3A interacted with full-length GRA18 (amino acids 26 – 648) but not with any of the GRA18 fragments tested. (B) Schematic representation of full-length (amino acids 26 – 648) and truncated versions of GRA18 (FH-GRA18Nt(aa 26-440) and FH-GRA18Ct(aa 233-648)) proteins stably expressed in T-Rex cells. (C) Cytoplasmic fractions from T-Rex cells presented in (B) were immunoprecipitated with anti-Flag antibodies and analyzed by immunoblotting. Untransfected T-Rex cells were used as a mock. (D) Schematic diagram summarizing the interaction domain mapping of GRA18 obtained from Y2H and biochemical approaches.

DOI: https://doi.org/10.7554/eLife.39887.005

reduced virulence phenotype in mice when challenged by intraperitoneal injection (*Figure 4D*). While assessing β-catenin amounts and subcellular localization we showed that β-catenin signals remained unaffected upon *T. gondii* infection of both human (HFF) and murine (L929) cells regardless of the *GRA18* status (*Figure 4E and F*, compare Pru *ku80* with *Δgra18*). Intriguingly, when parasites expressed high levels of GRA18, referred to as GRA18+++, β-catenin signals drastically

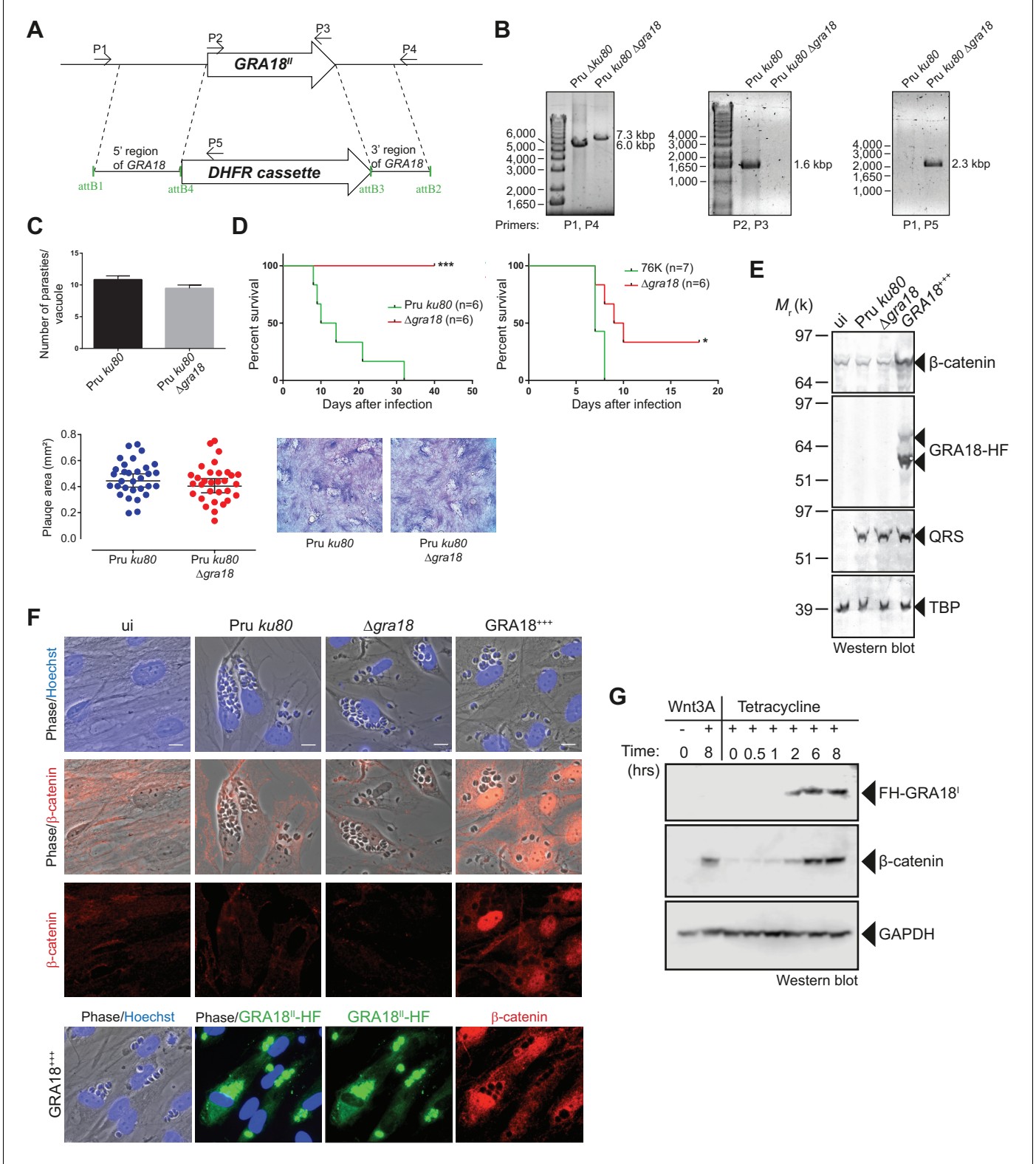

**Figure 4.** GRA18 is a positive regulator β-catenin. (**A**) Generation and confirmation of insertion/deletion of *GRA18* in *T. gondii* type II Pru strain. Schematic representation of the *GRA18* locus with the double homologous recombination event between the knockout construct (pDEST14 KO GRA18) and genomic DNA replacing the *GRA18* coding sequence with the *DHFR* cassette used for positive selection. (**B**) PCR reactions with the indicated primers confirming the deletion/insertion of *GRA18* in the mutant parasites. (**C**) Parasites lacking *GRA18* exhibit no growth defect in vitro as determined by fluorescence imaging assays in HFFs and plaque assays. Data are mean value ± s.d. of triplicates from two independent experiments. (**D**) GRA18

*Figure 4 continued on next page*

Figure 4 continued
mutants exhibit decreased virulence in mice. Virulence of the type II Pru *ku80Δgra18* and 76K *Δgra18* strains were compared to the parental strains Pru *ku80* and 76K, respectively, in BALB/c mice. Mice were inoculated with $10^5$ tachyzoites of each strain by intraperitoneal injection and survival was monitored. Cumulative results of two independent experiments with at least three mice in each group (n ≥ 6) are shown. Significance was tested using Log-rank (Mantel-Cox) test; *p=0.0162 and ***p=0.0005 when compared to the respective wild-type strain. (E) Effect of GRA18 on β-catenin levels. Murine L929 cells were left uninfected (ui) or infected with wild-type (Pru *ku80*), *Δgra18,* or the *Δgra18* GRA18$^{+++}$ complemented (GRA18$^{+++}$) strains. At 18 hr post-infection, cells were harvested and analyzed by immunoblot using the indicated antibodies. TgQRS was used to control parasite loading. (F) IFA of β-catenin in confluent HFFs left uninfected (ui) or infected with the indicated strains for 18 hr. In the lower panel, IFA was performed using an anti-HA antibody to monitor the HF-tagged version of GRA18 in the GRA18$^{+++}$ complemented strain. Data are representative of at least three independent experiments. (G) Immunoblot analysis of nuclear fraction of the T-Rex FH-GRA18 cell line left untreated or induced with tetracycline for the indicated periods of time. The Wnt3A ligand was used as a positive control.

DOI: https://doi.org/10.7554/eLife.39887.006

increased in cells infected by the GRA18$^{+++}$ strain compared to uninfected cells (*Figure 4E and F*). β-Catenin upregulation is associated with a strong accumulation of β-catenin in the host nuclei suggesting that a transcriptionally active bulk of β-catenin was produced (*Figure 4F*). To determine whether additional parasite factors may contribute to the GRA18-mediated β-catenin induction, we used ectopic expression of GRA18 in HEK293 human cell line. As a positive control, cells were treated with exogenous Wnt3A ligand, a natural inducer of β-catenin, thus confirming that the T-Rex cells carry a functional and regulatable β-catenin-destruction complex (*Figure 4G*) (*Azzolin et al., 2014*). Induction of GRA18 expression with tetracycline promoted the accumulation of β-catenin, indicating that GRA18 alone was sufficient to drive β-catenin upregulation to levels comparable to those obtained with Wnt3A (*Figure 4G*). Therefore, these data indicate that GRA18, very likely through the interactions with GSK3 and PP2A-B56, functions as a positive regulator of β-catenin.

## GRA18 alters the expression of a specific set of genes in infected cells

The ability of GRA18 to promote nuclear accumulation of the transcriptional regulator β-catenin prompted us to investigate whether GRA18-GSK3/β-catenin partnerships could contribute to the typical changes in gene expression observed in cells infected with tachyzoites. To test this hypothesis, we performed a comparative transcriptomic analysis by RNA-sequencing of mouse Bone Marrow-Derived Macrophages (BMDMs) loaded with parental or *Δgra18* parasites of the type II Pru strain. Macrophages were chosen because they are infected in mice and play an essential role in the early immune response against *T. gondii* (*Dunay et al., 2008*; *Jensen et al., 2011*). Since GSK3 and β-catenin were involved in the regulation of the inflammatory gene expression in response to bacterial lipopolysaccharides (LPS) (*Chattopadhyay et al., 2015*; *Jang et al., 2017*; *Martin et al., 2005*; *Yang et al., 2010*), transcriptomic analysis was also performed on infected macrophages treated with LPS. We focused our analysis on genes that were modulated with more than 3-fold change and had a signal threshold above 5 Reads Per Kilobase of transcript per Million mapped reads (RPKM) in at least one sample when comparing the wild-type and *Δgra18* mutant strains. Filtered data are presented in *Supplementary file 3* (data are accessible through NCBI GEO, accession number GSE103113). Thirty-eight genes were significantly and differentially regulated with most corresponding to genes up-regulated upon macrophage infection by wild-type parasites, but not with the *Δgra18* strain as revealed by hierarchical clustering (*Figure 5A*). Complementation of *Δgra18* mutation with GRA18-HF under the control of the *GRA1* promoter (*Δgra18,* GRA18$^{+++}$) restored the expression pattern to levels observed with wild-type parasites or even higher, in strong support of the GRA18-dependent induction of those genes. Parasite transcriptome analysis indicated that neither *Δgra18* mutation nor LPS treatment had any significant impact on *T. gondii* gene expression and that *GRA18* expression in the Pru *ku80 Δgra18,* GRA18$^{+++}$ complemented strain was restored with a ~ 35 fold more transcript reads than in the wild-type strain (*Figure 5D*, *Supplementary file 3*, and *Figure 5—figure supplement 1A*), in line with *GRA18* overexpression driven by the *GRA1* strong promoter. It is noteworthy that the expression pattern of the aforementioned host genes regulated by GRA18 remained similarly regulated in the infected macrophages subjected to LPS stimulation (*Figure 5A and B* and *Supplementary file 3*), indicating that LPS did not affect GRA18 function. Gene set enrichment analysis (GSEA) highlighted pathways significantly affected in a GRA18-dependent fashion (*p* values < 0.05) with functions related to the inflammatory response (e.

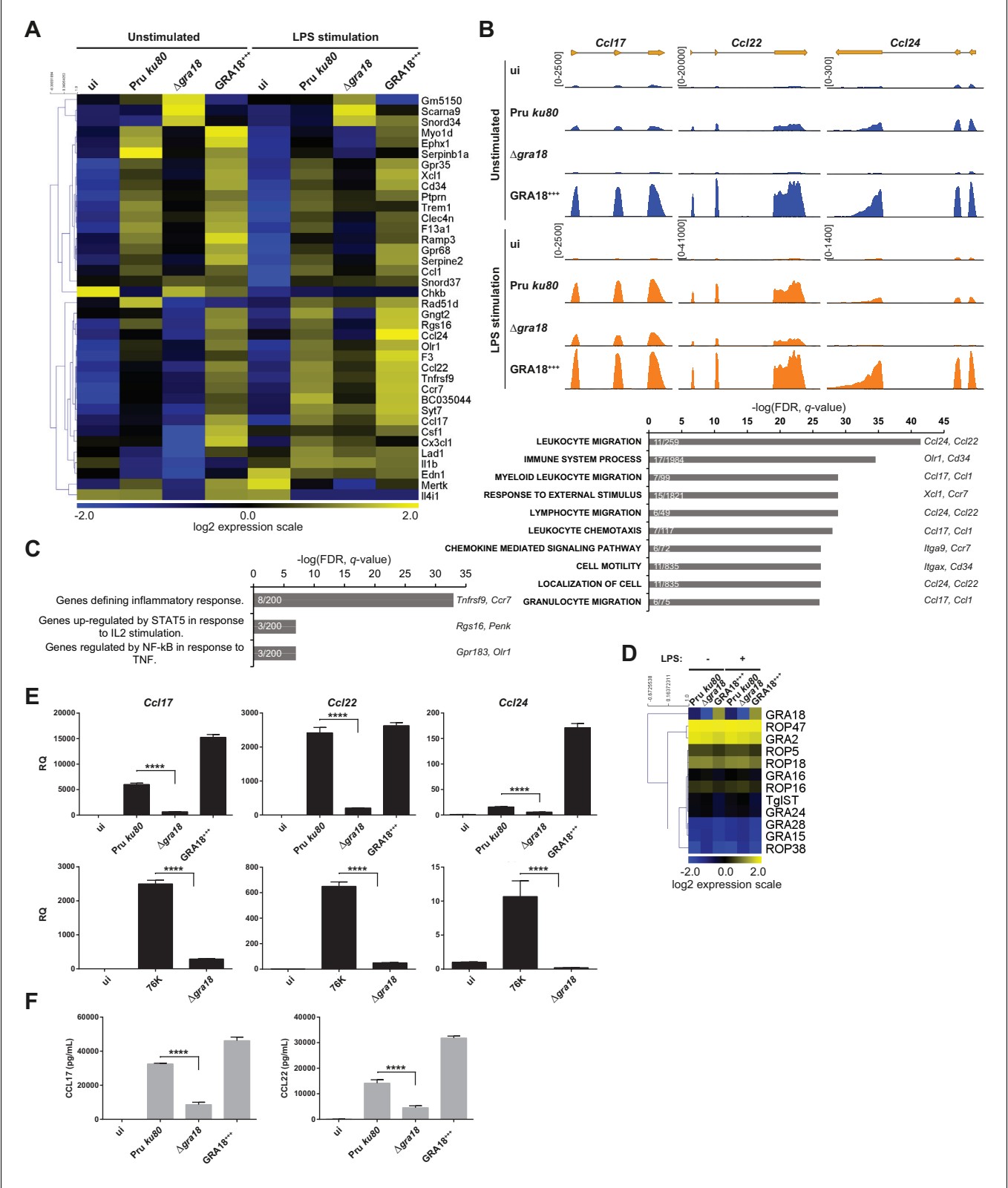

**Figure 5.** GRA18 alters the host cell transcriptome. (**A**) RNA-Seq analysis of BALB/c BMDMs that were left uninfected (ui) or infected with the indicated strains at an MOI of 1:5. At 18 hr post-infection, cells were left unstimulated or stimulated with LPS (100 ng/mL) for 6 hr. Heat map representation of the differentially expressed mouse genes (≥3 fold, RPKM ≥5 in at least one sample) between parental and Δ*gra18* infected cells in the absence of LPS. RPKM values were log2 transformed, Gene/Row normalized, and mean centered using MeV. (**B**) Tracks of the BMDMs RNA-Seq reads for *Ccl17, Ccl22, Figure 5 continued on next page*

Figure 5 continued

and *Ccl24* in the indicated samples. (C) Enrichment analysis in functional annotation and GO analysis of the differentially expressed genes defined in *Figure 5A* using GSEA. Top-scoring pathways regulated in a GRA18-dependent manner, number of genes per pathway, and names of representative genes are shown. (D) Heat map of expression values obtained by RNA-Seq analysis of *T. gondii* genes from the samples in (A). A selected set of *T. gondii* ROPs and GRAs are shown. (E) Quantitative chemokine expression was determined by qRT-PCR on BMDMs infected with the wild-type and Δ*gra18* mutant strains in the type II strains Pru and 76K. Values were normalized to the amount of TATA box binding protein (*Tbp*) in each sample. Data are mean value ± s.d. of three replicates. The *P*-values were calculated using two-tailed unpaired Student's *t*-test; ****p<0.001. Data are representative of two independent experiments. (F) CCL17 and CCL22 levels were measured by ELISA from supernatants collected 24 hr after BMDM infection with the indicated strains at an MOI of 1:5. Supernatant from uninfected cells was used as a control (ui). Means of four independent experiments are shown. ****, p<0.0001 following one-way ANOVA analysis and multiple-comparison *post hoc* tests.

DOI: https://doi.org/10.7554/eLife.39887.007

The following figure supplements are available for figure 5:

**Figure supplement 1.** RNA-Seq analysis of mouse and *T.*

DOI: https://doi.org/10.7554/eLife.39887.008

**Figure supplement 2.** Deletion of the aspartyl protease *ASP5* compromises GRA18 export and phenocopies the deletion of *GRA18* in promoting host cell gene expression.

DOI: https://doi.org/10.7554/eLife.39887.009

g., *Csf1*, *Olr1*, and *Tnfrsf9*) and chemotaxis (e.g., *Ccr7*, *Ccl1*, *Ccl17*, *Ccl22*, and *Ccl24*) (*Figure 5C*). Surprisingly, none of the typical Wnt/β-catenin target genes were found differentially regulated by *GRA18* (*Figure 5—figure supplement 1B* and *Supplementary file 3*), suggesting either that murine macrophages have a peculiar repertoire of β-catenin target genes or that GRA18 fosters a singular β-catenin transcriptional activity or that its action is independent of β-catenin.

To confirm the transcriptomic profile, we focused on the chemokines *Ccl17*, *Ccl22*, and *Ccl24* previously identified as selectively up-regulated genes upon *T. gondii* infection (*Hammoudi et al., 2015*; *Melo et al., 2013*). Quantitative RT-PCR analysis reproducibly demonstrated a pronounced decrease in *Ccl17*, *Ccl22*, and *Ccl24* expression from cells infected with Δ*gra18* mutant parasites when compared to wild-type Pru or Pru Δ*gra18*, GRA18+++ parasites (*Figure 5E*). Similar data were obtained with the 76K strain, another type II cystogenic strain, highlighting that *T. gondii* reproducibly regulated those chemokines in a *GRA18*-dependent manner. GRA18 clearly plays a major role in regulating CCL17 and CCL22 synthesis and secretion since its loss significantly lowered protein levels in the supernatant of mouse macrophages (*Figure 5F*). Overall, these results support the regulatory function of GRA18 for specific chemokines amongst others.

## GRA18 alters host gene expression in a β-catenin-dependent fashion

To evaluate whether β-catenin is required for the GRA18-induced chemokine expression, we generated macrophage RAW264.7-derived cell line mutated for *Ctnnb1* using CRISPR/CAS9-mediated gene editing. Cells were transfected with a vector expressing the CAS9 endonuclease and a guide RNA targeting the third coding exon of the murine *Ctnnb1* gene (*Figure 6A*). Cells harboring insertion/deletion (indel) mutations at the *Ctnnb1* locus lack β-catenin expression when compared to the wild-type parental cell line (*Figure 6A and B*). As expected, treatment of the *Ctnnb1*−/− mutant cells with the GSK3 inhibitors did not lead to β-catenin expression, otherwise induced in the wild-type parental cell line. These results demonstrate that the RAW264.7 murine cells carry a regulatable β-catenin destruction complex and that the two alleles coding for β-catenin were successfully disrupted in the *Ctnnb1*−/− mutant cells.

The ectopic expression of GRA18 in RAW264.7 macrophages caused a significant increase in endogenous β-catenin (*Figure 6C and D*, compare the effect of mCherry with GRA18 expression), similarly to the effect of GRA18 when delivered by the parasites in HFF or L929 cells (*Figure 4E and F*). As expected for RAW264.7 macrophages whose *Ctnnb1* alleles are disrupted, no β-catenin was detected upon ectopic expression of GRA18 (*Figure 6C and D*) in contrast to the significant amounts of nuclear β-catenin found in wild-type RAW264.7 macrophages expressing GRA18. In the same line *Ccl17*, *Ccl22*, and *Ccl24* were selectively induced in GRA18-expressing wild-type RAW264.7 macrophages when compared to mock - that is, mCherry - transfected cells (*Figure 6D and E*) whereas *Ccl17* and *Ccl24* were no longer induced in the β-catenin lacking ones. The slight reduction of *Ccl22* expression in the GRA18-expressing mutant RAW264.7 macrophages (*Figure 6E*)

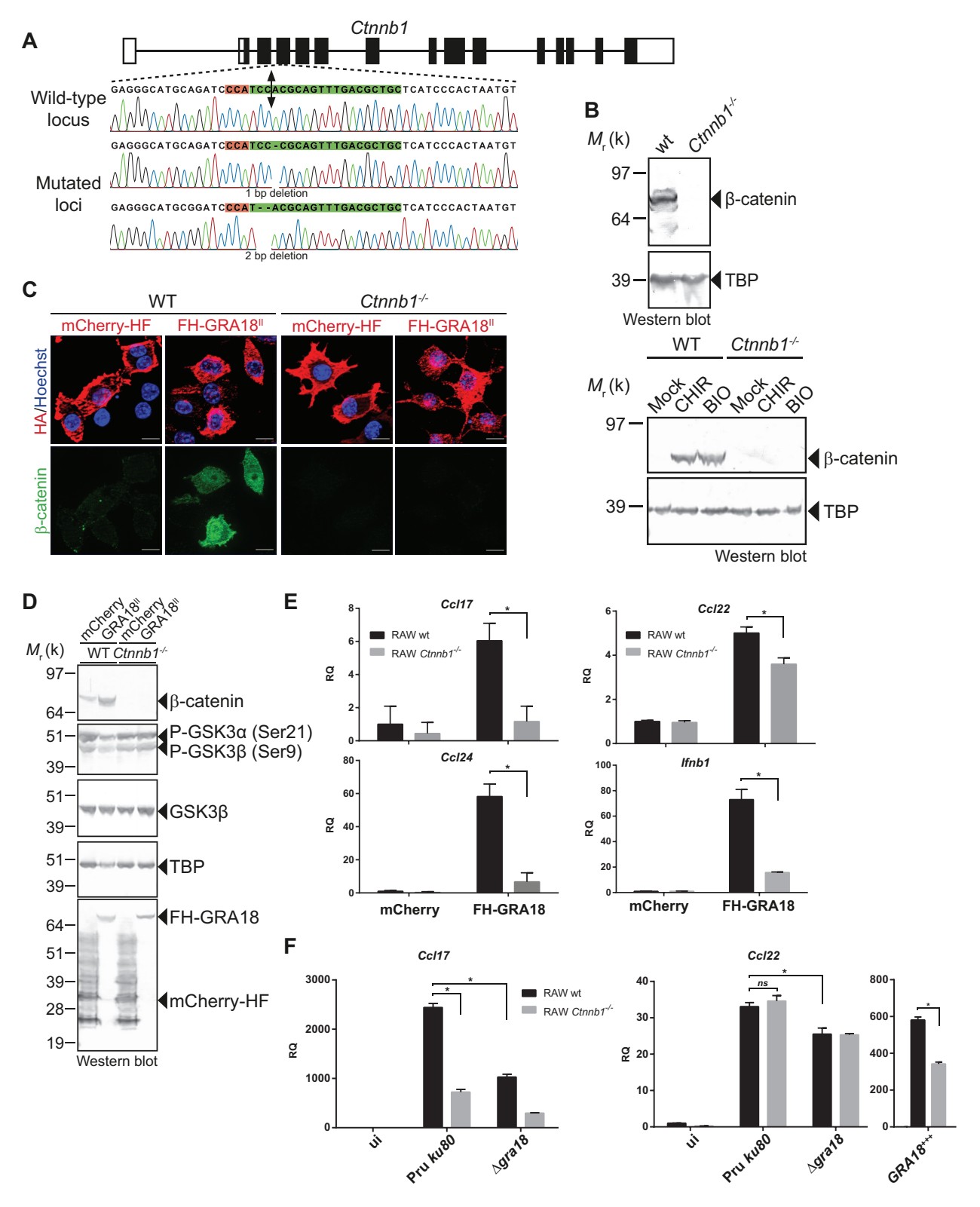

**Figure 6.** GRA18 promotes *Ccl17*, *Ccl22* and *Ccl24* chemokines expression in a β-catenin-dependent fashion. (**A**) Schematic diagram of gRNAs targeting the *Ctnnb1* locus. The Protospacer Adjacent Motif (PAM) sequence is lined and highlighted in red; the targeting sequences is shown in green. Bi-directional arrow indicates the Cas9 cleavage site. DNA sequences from the wild-type and mutated RAW264.7-derived cell lines were analyzed by DNA sequencing; the identified deletions are indicated with (-). No mutations in control samples were observed. (**B**) Immunoblot analysis of β-catenin in

*Figure 6 continued on next page*

*Figure 6 continued*

parental RAW264.7 and the *Ctnnb1^-/-^* mutant confirmed the absence of β-catenin expression. As positive controls, cells were treated with 3 mM of BIO or 2 mM of CHIR GSK3 inhibitors for 12 hr. TBP was used as loading control. (**C–E**) RAW264.7 (WT) and β-catenin-deficient (*Ctnnb1^-/-^*) RAW264.7-derived cell lines were transfected with mCherry vector control (pcDNA-mCherry-HF) or the FH-GRA18^II^ expression vector (pcDNA-FH-GRA18^FL^). At 18 hr after transfection, cells were either (**C**) fixed for IFA using anti-HA (red) and anti-β-catenin (green) antibodies or (**D**) cells were harvested and analyzed by immunoblot using the indicated antibodies. Anti-HA was used to detect FH-GRA18 and mCherry-HF. In (**E**) and (**F**) transcripts for *Ccl17, Ccl22, Ccl24,* and *Ifnb1* were quantified by qPCR and normalized using *Tbp*. Data are mean value ± s.d. of three replicates. The *P*-values were calculated using two-tailed unpaired Student's *t*-test or one-way ANOVA with Bonferroni posttests analysis of variance; *p<0.05 and *P* values greater than 0.05 were considered not significant (*ns*). In (**F**) RAW264.7 (WT) and β-catenin-deficient (*Ctnnb1^-/-^*) RAW264.7-derived cell lines were either left uninfected or infected with the indicated wild-type and Δ*gra18* mutant strains at a MOI of 1:6 for 18 hr. Data are representative of two independent experiments.
DOI: https://doi.org/10.7554/eLife.39887.010

suggests a different mode of regulation for the three chemokines. Consistent with β-catenin upregulation, similar data were obtained with *Ifnb1* (*Figure 6E*), a known β-catenin target gene in macrophages (*Ma et al., 2014*; *Rathinam et al., 2010*; *Yang et al., 2010*). To ascertain the β-catenin requirement for GRA18 activity under more physiological conditions, chemokine expression was monitored upon infection of RAW264.7 cells disrupted or not for *Ctnnb1* with wild-type and Δ*gra18* mutant parasites. While β-catenin was required for full induction of the *Ccl17* chemokine, *Ccl22* expression remained unaffected in *Ctnnb1^-/-^* mutant host cells (*Figure 6F*), again consistent with a different mode of regulation orchestrating *Ccl17* and *Ccl22* expression. Importantly, the moderate role played by GRA18 in the regulation of *Ccl22* in RAW264.7 cells when compared to BMDMs, suggests that a GRA18-independent pathway controls *Ccl22* expression, which is reminiscent of the aforementioned β-catenin independent mechanism (see discussion). Indeed, *Ccl22* was much less induced in RAW264.7 than in BMDM cells (compare *Figures 5E* and *6F*), indicating differences in responsiveness to *T. gondii* infection. In addition, these data also argue that using the transformed RAW264.7 cell line to monitor GRA18 activity on *Ccl22* induction may be less relevant than relying on the primary BMDM cells. Altogether, these results indicate that basal levels of β-catenin are required for full activity of GRA18 towards *Ccl17*.

## GRA18 acts as an inhibitor of GSK3 to trigger β-catenin transcriptional activity

In order to better map the GRA18 active domains we assayed the different sub-domains of GRA18 (GRA18^Nt^ and GRA18^Ct^) described above for their ability to upregulate β-catenin and to induce expression of the downstream target genes in macrophages. While expression of GRA18^FL^ in RAW264.7 macrophages caused a significant increase in endogenous β-catenin protein levels, leading to the expression of the downstream chemokines (*Figure 7A and B*), expression of GRA18^Nt^, which no longer interacts with GSK3 and PP2A-B56 (*Figure 3*), failed to induce β-catenin and accordingly chemokine transcripts. Given that the N-terminal region of GRA18 has retained the ability to interact with β-catenin (*Figure 3A*), these data suggest that the interactions with β-catenin alone are not sufficient to promote its accumulation and transcriptional activity. In contrast, GRA18^Ct^, shown previously to interact with GSK3 and PP2A-B56 (*Figure 3*) has retained full activity on β-catenin and downstream transcriptional consequences, further confirming the pivotal role of β-catenin in driving the expression of the aforementioned chemokines. While the effect of GRA18 on β-catenin protein levels required interactions with GSK3 and/or PP2A-B56, direct interaction between GRA18 and β-catenin seems to be dispensable. Altogether these results support a scenario in which GRA18, through its interactions with GSK3 and PP2A-B56 inhibits the β-catenin-destruction complex, hence stabilizing β-catenin thereby acting as a positive regulator of β-catenin level to promote gene expression. Additional evidence for this model is the expression of *Ccl22* and *Ccl24* induced by GSK3 inhibitors in a manner that resembles the effect of GRA18 (*Figure 7C and D*).

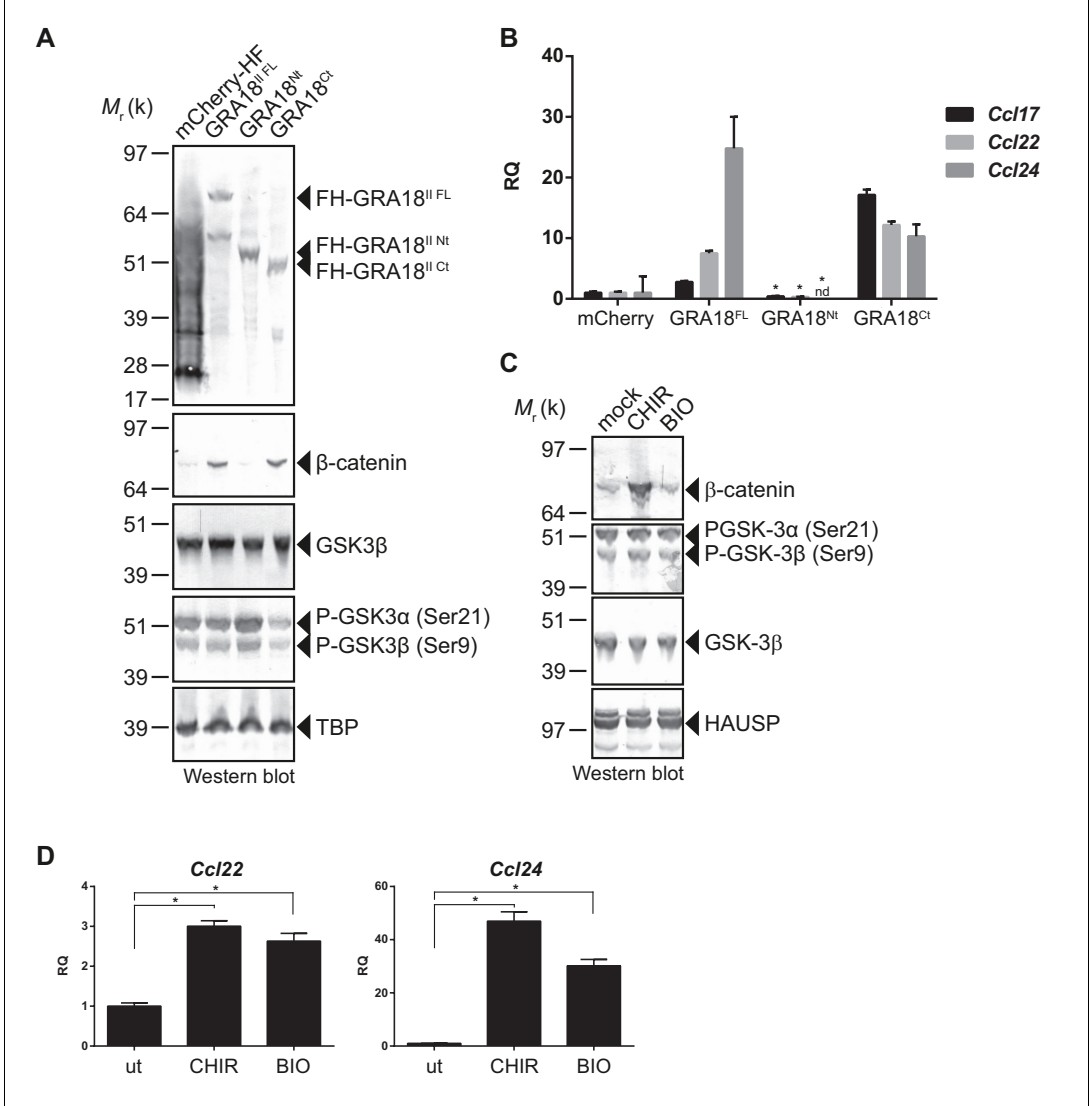

**Figure 7.** GRA18 activity is dependent on its interactions with GSK3 and PP2A-B56. (A–B) RAW264.7 cells were transfected with mCherry vector control or the FH-GRA18 expression vectors pcDNA-FH-GRA18$^{FL}$ (GRA18$^{FL}$), pcDNA-FH-GRA18$^{Nt}$ (GRA18$^{Nt}$), and pcDNA-FH-GRA18$^{Ct}$ (GRA18$^{Ct}$). At 18 hr after transfection, cells were harvested and (A) whole cell extracts were analyzed by immunoblot using the indicated antibodies, or (B) quantitative chemokine expression was determined by qRT-PCR as in *Figure 6*. Asterisks indicate *P*-values (p<0.05) obtained when comparing GRA18$^{Nt}$ with GRA18$^{FL}$. (C) Treatment of RAW264.7 cells with the GSK3 inhibitors CHIR and BIO led to β-catenin protein accumulation and *Ccl22* and *Ccl24* upregulation as determined by qRT-PCR in (D). ut, untreated; nd, not detected. Data are mean value ± s.d. of three replicates. The *P*-values were calculated using two-tailed unpaired Student's *t*-test; *p<0.05. Each data set is representative of two independent experiments.
DOI: https://doi.org/10.7554/eLife.39887.011

## Discussion

### GRA18 export mechanism

In search of new *T. gondii* effectors from the GRA family that would be not only secreted into the PV space but would also traffic beyond - that is, across the PVM - and be delivered in the infected host cell, we identified a novel *T. gondii* GRA protein that we termed GRA18. However, contrasting with the recently described GRA16, GRA24, GRA28 or TgIST, whose final destination is the nucleus of the host cell, GRA18 remained strictly cytoplasmic throughout the intracellular life cycle of the tachyzoite.

Analyzing GRA18 protein sequence, we recognized in addition to the N-terminal signal peptide, an N-terminal putative TEXEL motif, both embedded in highly intrinsically disordered regions (*Figure 1E*). The presence of the TEXEL motif suggests that GRA18 might be processed by the aspartic protease TgASP5, an enzyme we and others have recently shown to mediate the secretion or export of a number of GRA proteins (*Coffey et al., 2015*; *Curt-Varesano et al., 2016*; *Hammoudi et al., 2015*). As expected for a TgASP5-dependent export, GRA18 was no longer detected in the cytoplasm of cells invaded by *TgASP5* mutant parasites (*Figure 5—figure supplement 2A*) and accordingly, *TgASP5* and GRA18 mutant parasites showed similar inability to induce target genes in macrophages ((*Hammoudi et al., 2015*), *Figure 5—figure supplement 2B*). Recent findings showed that the export pathway to the host cell of most of the dense granule effectors depends on MYR1, another DG protein localized at the PVM. In agreement with a common GRA export mechanism, trafficking of GRA18 through the PVM was also dependent on MYR1 (*Figure 1D*), (*Coffey et al., 2016*). The detection of GRA18 in the host cell cytoplasm together with GRA18 sequence features prompted us to search for host cell cytoplasmic partners.

## GRA18, a potential inhibitor of the β-catenin destruction complex

Using a genetic screen combined with affinity- and size-exclusion chromatography under stringent conditions, and mass spectrometry, we have characterized GRA18 as a strong interactor of host GSK3 and the PP2A-B56 holoenzyme, and to a lesser extent with β-catenin. The complex(es) promote β-catenin nuclear accumulation, raising the possibility that GRA18 could modulate β-catenin stability (see proposed model in *Figure 8*). However, while GRA18 clearly regulated host cell gene expression, we did not detect significant changes in β-catenin levels upon infection with wild-type parasites (*Figure 4E and D*). As only a small fraction of the total β-catenin pool is destined to transcriptional regulation, possibly GRA18 induces subtle effects on the β-catenin levels that cannot be captured by IFA or immunoblot. Next, to figure out how GRA18 could interplay with β-catenin in cells, we engineered β-catenin mutant in RAW264.7 macrophages and provided definitive evidence for β-catenin acting epistatically to GRA18's activity on host cell transcription for a restricted set of target genes (i.e. *Ccl17, Ccl24,* and *Ifnb1*) (*Figure 6*).

β-Catenin is a dual function co-activator protein with distinct pools serving cell-cell adhesion and gene transcription functions, respectively. β-Catenin was originally identified as an element of the adherent junction complex together with cadherin and α-catenin, the latter binding to actin filaments in a process that promotes cell–cell contacts. Further, a nonjunctional pool of cytoplasmic β-catenin was recognized as a transcriptional coactivator of Wnt canonical signaling pathway. The Wnt signaling represents a complex and essential pathway starting with a repertoire of Wnt ligands and acting at the heart of embryogenesis, but also throughout life by directing stem cell renewal and senescence. Centerpiece of Wnt signaling is the canonical Wnt/β-catenin pathway genetically dissected in both human and *Drosophila*. Briefly, in the absence of Wnt ligands, cytosolic β-catenin is maintained at low levels by the multiprotein complex composed of APC, Axin, GSK3 and PP2A-B56. Within this complex, also referred to as the β-catenin destruction complex, β-catenin is constantly phosphorylated for ubiquitylation that directs its degradation by the 26S proteasome (*Liu et al., 2002*). Mutation in any of these components (APC, Axin, or β-catenin) leads to inappropriate stabilization of β-catenin, which results in cancer, most notably of the colon (*Stamos and Weis, 2013*). Axin, a central scaffold protein of the complex, is typified by intrinsically disordered and flexible regions that mediate direct interactions with all other core components of the destruction complex (β-catenin, APC, and GSK3) (*Stamos and Weis, 2013*). Remarkably, the *T. gondii* tachyzoite GRA18 also carries significant disordered regions (*Figure 1E*) which could confer the potential of competing with Axin/APC for binding to β-catenin, GSK3, and PP2A-B56. Accordingly, by disrupting the β-catenin destruction complex GRA18 would likely promote the stabilization and nuclear translocation of β-catenin and ultimately β-catenin-dependent gene expression (*Figure 8*). RNA-Seq performed on BMDMs indicated that β-catenin transcripts remained unaffected by GRA18 (*Supplementary file 3*), which is consistent with a post-transcriptional regulation of β-catenin. Collectively our data argue for GRA18 acting as a direct regulator of β-catenin and although the exact *modus operandi* of GRA18 awaits clarification, some scenarios can be proposed. Interestingly, the Y2H interaction assay allowed defining the region of β-catenin that promotes the interaction with GRA18 (*Figure 2A*). Indeed, the SID analysis for interaction between GRA18 and β-catenin is mediated by a region encompassing amino acids 42 – 297 of the latter (*Figure 2A*), which harbors conserved serine and threonine

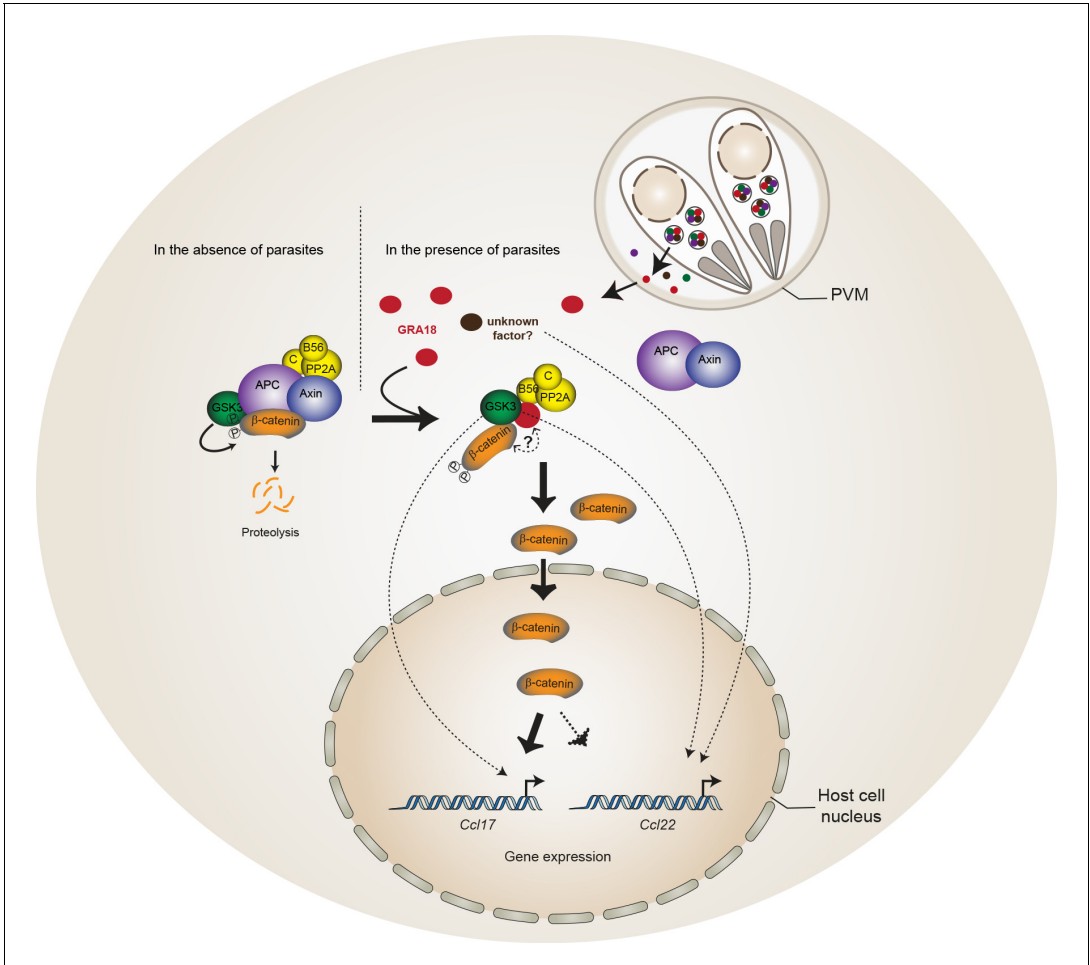

**Figure 8.** Model for GRA18 mechanism of action. A model depicting how GRA18 interacts and interferes with the β-catenin destruction complex leading to host cell gene regulation. Possible effect of GRA18 on gene expression through GSK3 but β-catenin-independent or still unidentified *T. gondii* factor are represented by dashed lines. Question mark indicates the putative direct interaction with β-catenin observed by the Y2H or GRA18 overexpression. See also discussion.

DOI: https://doi.org/10.7554/eLife.39887.012

residues initially phosphorylated by CK1 (Ser45), and subsequently by GSK3 (Thr41, Ser33 and Ser37). Once phosphorylated, Ser33 and Ser37 mediate the interaction with the β-TrCP adaptor protein for β-catenin degradation. Therefore, the GRA18 binding features to the N-terminal domain of β-catenin appear quite peculiar and differ from the binding sites for Axin and APC mapped at central ARM domain. As such, the binding interface of GRA18 to the β-catenin N-terminal region would confer to GRA18 a unique ability to interfere with both the phosphorylation cascade catalyzed by CK1 and GSK3, but also at a later step with the binding to β-TrCP. Interestingly, phosphorylation of GSK3β Ser9 or GSK3α Ser21 is reported to decrease GSK3α/β enzymatic activity (*Cross et al., 1995*; *Peyrollier et al., 2000*). Because GRA18 did not alter the phospho-Ser21/Ser9 levels (*Figures 6D* and *7A*), we conclude that the parasite protein, which we found to bind GSK3 and inhibit its pro-degradative activity on cytoplasmic β-catenin, acts independently of the phosphorylation status of Ser21/Ser9.

In the context of the Wnt signaling, the function of the PP2A-B56 when associated with the β-catenin destruction complex remains uncertain (*Hsu et al., 1999*; *Seeling et al., 1999*; *Yamamoto et al., 2001*). A recent study reported that the PP2A regulatory B56 subunit binds to a LxxIxE Short Linear Motif (SLiM) on partner proteins as illustrated with the B56 binding to Axin (*Hertz et al., 2016*). Interestingly, GRA18 carries two LxxIxE motifs within an intrinsically disordered region of the C-terminal domain, which is in agreement with the B56 binding assays (*Figure 3A and*

*C*). As we previously showed (*Hakimi et al., 2017*), SLiMs is a prominent characteristic of the GRA effector family of proteins, probably reflecting a favorable evolutionary strategy to expand and diversify the binding interfaces between host and parasite proteins.

Our data suggested a simultaneous binding of GRA18 to GSK-3 and PP2A-B56 (*Figure 2E*), but whether the binding of PP2A-B56 is essential for GRA18 activity is yet to determine. GSK3 activity can be regulated by phosphorylation of tyrosine residues (Tyr279/Tyr216 in α/β isoforms, respectively) exposed in the activation loop, which enhances its kinase activity (*Dajani et al., 2003*). An attractive hypothesis is that PP2A-B56 is recruited to the GRA18 complex in order to dephosphorylate GSK3, hence contributing to the inhibitory activity towards the β-catenin destruction complex. Alternatively, PP2A-B56 could be recruited to GRA18-GSK3 complex to dephosphorylate β-catenin, leading to the inhibition of β-catenin degradation (*Su et al., 2008*).

## The GRA18-GSK3-β-catenin axis induces the expression of anti-inflammatory chemokines

Our transcriptomic data have revealed the magnitude by which GRA18 alters the expression of genes of cells infected by *T. gondii* tachyzoites, particularly the chemokines *Ccl17*, *Ccl22*, and *Ccl24*. Those genes have been found repeatedly induced by *T. gondii* tachyzoites regardless of the strain type in murine macrophages (*Figure 5—figure supplement 1C*) (*Hammoudi et al., 2015*; *Melo et al., 2013*; *Morgado et al., 2011*), as expected from the overall conservation of GRA18 across the *Toxoplasma* lineages. Whether this GRA18 property is also expressed in other intermediate hosts or in different cell types is not known, but given the conservation of the Wnt signaling in vertebrates, GRA18 activity in other hosts can be assumed. Interestingly, the human placental cells, which have the unique ability to resist to *T. gondii* infection, were shown to specifically respond to the invader by producing the CCL22 chemokine (*Ander et al., 2018*), whereas other cell types such as human fibroblasts did not ((*Ander et al., 2018*) and data not shown). Whether the mechanism underlying CCL22 induction in human placental cells is driven by GRA18 or differ from murine macrophages should be addressed in the future. It is noteworthy that a quite substantial residual activation of *Ccl17*, and *Ccl22* was observed in the absence of GRA18 (*Figures 5E–F and* and *6F*), which suggests that an alternative mechanism of activation by the parasites exists.

We provide here genetic evidence for GRA18 acting in a β-catenin-dependent manner on chemokine expression in macrophages. Although Wnt β-catenin signaling has originally been studied in the context of thymocyte development and stem cell biology (*Reya et al., 2003*; *Staal et al., 2008*; *Verbeek et al., 1995*), there is increasing evidence for its contribution in the regulation of innate immunity. For instance, activation of β-catenin promotes differentiation and activation of dendritic cells to stimulate regulatory T cells (Tregs) and suppresses the inflammatory response (*Manicassamy et al., 2010*; *Zhou et al., 2009*), while β-catenin accumulation triggered by GSK3 inhibitors enables Treg cell survival (*Ding et al., 2008*). Interestingly, CCL17, CCL22 and CCL24 are expressed by alternatively activated M2-polarized macrophages or tolerant macrophages and the release of these chemokines results in the recruitment of Treg cells and amplification of a Th2 response (*Biswas and Mantovani, 2010*). Thereby, an attractive hypothesis could be that once released in the cell cytoplasm, GRA18 contributes to the characteristic M2 macrophage polarization associated with the *T. gondii* parasitic process (*Jensen et al., 2011*) through the GSK3-β-catenin axis. It is well acknowledged that cytokines and chemokines act as a frontline defense against *T. gondii* since they promote rapid Th1 cell–mediated pro-inflammatory response typified by the recruitment of immune killer cells at play to control the bulk tachyzoite population. However, counterbalancing the Th1-induced inflammatory effects, Th2 chemokines are efficient at dampening the inflammatory response and are therefore crucial to avoid immunopathology and preserve host and parasite survival. Accordingly these Th2 chemokines likely favour parasite dissemination and colonization of deep organs such as the brain allowing *T. gondii* bradyzoite to subsequently establish a persistent parasitism within intracellular cysts. Consequently, we propose the following hypothesis in which the infection using Δ*gra18* strains, which may provide a weaker Th2 chemokine profile (*Ccl17*, *Ccl22*, and *Ccl24*) led to early control and lower chance for long-term persistence in mice than *GRA18* wild type parasites.

In conclusion, the discovery and characterization of GRA18 extends our view of how *T. gondii* has evolved a variety of strategies to interfere with host cells by unveiling the hijacking of the evolutionarily conserved β-catenin destruction complex.

## Materials and methods

### Parasite and cell culture

*T. gondii* tachyzoites were maintained by serial passage on human foreskin fibroblast (HFF) monolayers. The strains used in this study were RH *ku80*, Pru *ku80*, and 76K-GFP-LUC (gift of M. Grigg, National Institutes of Health, Bethesda, MD). HFF primary cells, T-Rex-293 (RRID:CVCL_D585), L929 (Sigma-Aldrich Cat# 85011425), J774 (J774A.1, Sigma-Aldrich Cat# 91051511), and RAW264.7 (ATCC Cat# TIB-71, RRID:CVCL_0493) cell lines were cultured in Dulbecco's modified Eagle's medium (DMEM, Invitrogen) supplemented with 10% heat-inactivated FBS (Invitrogen), 10 mM Hepes buffer, pH 7.2, 2 mM L-glutamine, and 50 µg/ml penicillin and streptomycin (Thermo Fisher Scientific). Cells were incubated at 37°C in 5% $CO_2$. Stable transgenic parasites or recombinants were selected with 25 µg/ml mycophenolic acid and 50 µg/ml xanthine or 1 µM pyrimethamine. The cultures were free of mycoplasma, as determined by qualitative PCR and/or IFA.

### Reagents

Antibodies raised against Hemagglutinin (Roche Cat# 3F10, RRID:AB_2314622 or Cell Signaling Technology Cat# 3724, RRID:AB_1549585), GRA1 (provided by J.-F- Dubremetz, UMR 5235 Centre National de la Recherche Scientifique, Montpellier, France), Toxofilin (provided by I. Tardieux, INSERM 1209, Grenoble, France), MIC2 (provided by D.Sibley, Washington University School of Medicine, St. Louis, MO), β-catenin (BD Biosciences Cat# 610153, RRID:AB_397554), GSK3β (Cell Signaling Technology Cat# 12456, RRID:AB_2636978), PP2A-B56 (#MABS270, Millipore), PP2A65 RA (Cell Signaling Technology Cat# 2039S, RRID:AB_10695607), PP2Ac (#2038, Cell signaling), Phospho-GSK3α/β (Cell Signaling Technology Cat# 8566S, RRID:AB_10860069), TBP (Abcam Cat# ab62126, RRID:AB_2287049), and TgQRS (*van Rooyen et al., 2014*) were used in the immunofluorescence assay and/or Western blotting. Immunofluorescence secondary antibodies were conjugated to Alexa Fluor 488 or Alexa Fluor 594 (Invitrogen). Western blotting secondary antibodies conjugated to alkaline phosphatase was purchased from Promega. The inhibitors CHIR 99021(#252917-06-9) and BIO (#66746362–9) were purchased from R and D systems. Recombinant bacterial lipopolysaccharide (LPS) from *Escherichia coli* O26:B6 (Sigma-Aldrich) was used to stimulate the BMDMs.

### Plasmid constructs

The plasmids and primers used in this work are listed in *Supplementary file 1*. To construct the vectors pLIC-GRA18-HA³-DHFR and pLIC-GRA18-HF-DHFR, the coding sequence of *GRA18* was amplified using primers LICF-288840_F and LICF-288840_R and RH *ku80* genomic DNA as the template. The PCR product was cloned to pLIC-HA³-*DHFR* and pLIC-HF-*DHFR* vectors, respectively using the LIC cloning method as described in *Huynh and Carruthers, 2009*.

The vector pDEST14 KO *GRA18^{II}* was generated to construct the deletion/insertion mutation of *GRA18* in Pru (type II) *T. gondii* strain. The Multisite Gateway Pro3-fragment Recombination system was used to clone the *DHFR* cassette flanked by the 5' and 3' surrounding regions of GRA18 coding sequence of type II genomic DNA as described in *Bougdour et al. (2013)*. Briefly, the 5' flanking region of *GRA18* of Pru strain was amplified using primer attB1-288840_F and attB4-288840_R, and was cloned into the plasmid pDONR221 P1-P4 (Invitrogen). The 3' flanking region of *GRA18* was amplified using primers attB3-288840_F and attB2-288840_R and was cloned into the plasmid pDONR221 P3-P2. The resulting vectors, pDONR221/5'*GRA18* and pDONR221/3'*GRA18*, respectively, were then recombined with pDONR221/*DHFR* into the destination vector pDEST14 KO, yielding the pDEST14 KO *GRA18^{II}*.

The plasmid pTOXO_Cas9-CRISPR::sgGRA18 vector was generated as previously described (*Curt-Varesano et al., 2016*) to construct the *gra18* deletion in the 76K strain. Briefly, the sense and anti-sense oligos GRA18-CRISPR-FWD and GRA18-CRISPR-REV containing the sgRNA targeting the *GRA18* genomic sequence were phosphorylated, annealed and ligated in the pTOXO_Cas9-CRISPR plasmid linearized with BsaI, yielding pTOXO_Cas9-CRISPR::sgGRA18.

To construct the pP_{GRA1}-GRA18^{II}-HF vector, the promoter sequence of *GRA1* was amplified by PCR using the primers LICF-P_{GRA1}_F2 and P_{GRA1}_R. The GRA18 coding sequence of type II *T. gondii* strain was amplified by PCR using the primers GRA1-GRA18^{1-x}_F and LICR-288840_R. The

resulting PCR products were assembled using the Gibson assembly kit (NEB) and the resulting P$_{GRA1}$-GRA18 DNA fragment was cloned into the plasmid pLIC-HF-*DHFR*, yielding the pPP$_{GRA1}$-GRA18$^{II}$-HF vector.

The pX330_hSpCas9::sgCTNNB1 vector was generated using the sense and anti-sense oligos, CTNNB1-CRISPR-FWD and CTNNB1-CRISP-REV, respectively. Annealed oligos were ligated into the pX330_hSpCas9 plasmid linearized with BbsI (*Cong et al., 2013*). The expressed guide RNA targets the second armadillo repeat of genomic *Ctnnb1* sequence.

## Toxoplasma transfection and generation of GRA18 mutant strains

*T. gondii* strains were transfected using electroporation parameters established previously (*Bougdour et al., 2013*). Stable integrants or recombinants were selected with 25 μg/ml mycophenolic acid and 50 μg/ml xanthine or 1 μM pyrimethamine, and cloned by limiting dilution.

To construct the deletion/insertion mutation of *GRA18* the type II Pru *ku80* strain, the pDEST14 KO *GRA18*$^{II}$ plasmid was amplified by PCR using primers attB1-288840_F and attB2-288840_R. After sodium acetate/ethanol precipitation, DNA was re-suspended in TlowE buffer and ~20 μg of PCR product was used for transfection. Recombinants were selected with 1 μM pyrimethamine and single-cell cloned by limiting dilution and verified by PCR analysis as described in *Figure 4*.

To generate the *GRA18* insertional mutant in the 76K strain, the parasites were cotransfected with a mixture of the pTOXO_Cas9CRISPR::sgGRA18 vector with purified amplicons containing the *DHFR* cassette flanked by sequences homologous to the sequence targeted by sgGRA18 (5:1 mass ratio). These amplicons were generated by PCR amplification of the *DHFR* cassette using the primers GRA18-DHFR - F and GRA18-DHFR – R, and a vector carrying the *DHFR* cassette as template (*Donald and Roos, 1993*). Stable recombinants were selected with 1 μM pyrimethamine, single-cell cloned by limiting dilution and verified by PCR analysis as described in *Figure 4*.

## Immunofluorescence microscopy

Immunofluorescence assays were performed previously (*Bougdour et al., 2013*). Briefly, cells were fixed in PBS-3% (vol/vol) formaldehyde and permeabilized with PBS-0.1% Triton X-100 (vol/vol) or ethanol (−20°C) for 3 min. After blocking in PBS-3% BSA, samples were incubated in PBS-3% BSA containing the primary antibodies indicated in the figures, followed by the secondary antibodies coupled with Alexa Fluor 488 or Alexa Fluor 568 (Invitrogen) at a 1:1000 dilution. Nuclei of both host-cells and parasites were stained for 10 min at RT with Hoechst 33258 at 2 μg/ mL in PBS. After four washes in PBS, coverslips were mounted on a glass slide with Mowiol mounting medium. Images were acquired with a fluorescence microscope AxioImager M2 equipped with Apotome module (Carl Zeiss, Inc.).

## Mice and experimental infection

Six-week-old female BALB/cJRj mice were obtained from Janvier Labs and were maintained in specific pathogen-free conditions in accordance with institutional and national regulations. Freshly egressed tachyzoites were washed and diluted in Hank's Balanced Salt Solution (HBSS) supplemented with 10 mM HEPES at pH7.2. Plaque assays were performed on each inoculum to quantify the number of viable tachyzoites injected, and only experiments where comparable numbers were obtained were included in our analyses. Mice were infected by intraperitoneal injection of 10$^5$ tachyzoites in 200 μL volume. The health of the mice was monitored daily until they presented severe symptoms of acute toxoplasmosis (bristled hair and complete prostration associated with reduced mobility). All animal experiments were conducted with the approval and oversight of the Institutional Animal Care and Use Committee at the University Grenoble Alpes (agreement # B38 516 10 006).

## Yeast two-hybrid screen

Full-length GRA18$^{II}$ (aa 27 to 648) cloned in pB27 (N-LexA-bait-C fusion) was used in a ULTImate Y2H screen against a human Human Placenta_RP5 complementary DNA Gal4-activating domain-fusion library (Hybrigenics, Paris, France). The construct was checked by sequencing. Prey fragments of positive clones were amplified by PCR and sequenced at their 5′ and 3′ junctions and the resulting sequences were used to identify corresponding interacting proteins in the GenBank database via a

fully automated procedure. A confidence score (Global Predicted Biological Score, Global PBS) was attributed to each interaction as previously described (*Formstecher et al., 2005*).

The interaction domain mapping was performed using Full-length GRA18$^{II}$ (aa 27 to 648) and GRA18 fragments (aa 26 to 233, aa 233 to 440, aa 420 to 648, aa 152 to 350, and aa 350 to 585) that were PCR-amplified and cloned in frame with the LexA DNA binding domain (DBD) by Gaprepair into plasmid pB27, derived from the original pBTM116 (*Vojtek and Hollenberg, 1995*). The Hybrigenics reference for this construct is hgx3694v1_pB27. This construct was used previously to screen the placenta library and was used as a control in this assay. The prey fragments for the human *CTNBB1* var3, *GSK3A* and *PPP2R5A* were extracted from the above ULTImate Y2H screening of GRA18 against the placenta cDNA library. The prey fragments were cloned in frame with the Gal4 Activation Domain (AD) into plasmid pP6, derived from the original pGADGH (*Bartel et al., 1993*). The AD constructs were checked by sequencing. The pP7 prey plasmid used as control in this assay is derived from the pP6 plasmid. Bait and prey constructs were transformed in the yeast diploid cells, obtained using a mating protocol with both yeast strains L40ΔGal4 (mata) and YHGX13 (Y187 ade2-101::loxP-kanMX-loxP, matα) (*Fromont-Racine et al., 1997*). In order to identify GRA18 fragments interacting with the three tested preys, three diploids for each fragment were picked and the interaction tested by growth assay (robot calibrated drops) on selective medium without tryptophan, leucine and histidine (DO-3). The interaction assays are based on the reporter gene HIS3 (growth assay without histidine). Five different concentrations of 3-AminoTriazol (3-AT), an inhibitor of the HIS3 gene product, were added to the DO-3 plates to increase stringency. The following 3-AT concentrations were tested: 1, 5, 10, 20 and 50 mM 3-AT.

## BMDM

Bone marrow derived macrophages were generated as described previously (*Bougdour et al., 2013*) with the following modifications. Briefly, bone marrow was flushed from femurs and tibias of BALB/c mice and cultured for 1 week in complete macrophage medium (Dulbecco modified Eagle's minimal essential medium (DMEM) (Invitrogen, Breda, the Netherlands) supplemented with 10% fetal calf serum (FCS) (Invitrogen), 20% conditioned medium from macrophage-colony stimulating factor-secreting L929 fibroblasts (*Aziz et al., 2009*), 50 μM 2-mercaptoethanol, 1X non-essential amino acids (Thermo Fisher Scientific), and 2% penicillin/streptomycin-glutamine. After 7–10 days in culture adherent cells were approximately 95% pure macrophages (F4/80$^+$) and cells were used for experiments.

## Toxoplasma growth assay by fluorescence imaging assays

*T. gondii* proliferation was determined by high-content fluorescence imaging as described previously (*Palencia et al., 2017*). Briefly, 10,000 HFFs were seeded in each well of well in 96-well plate and then infected with $4 \times 10^4$ parasites. Invasion was synchronized by briefly centrifuging the plate at 400 rpm, and plates were incubated at 37°C for 2 hr. Infected cells were then washed three times with PBS and further incubated for a total of 24 hr of growth. Nuclei were stained with Hoechst 33342 at 5 μg/mL for 20 min and after fixation and permeabilization, parasite vacuoles were labelled using an anti-GRA1 antibody (specific to *Toxoplasma*). Images were automatically acquired using the ScanR microscope system (Olympus) to determine the number of tachyzoites per vacuole. The experiments were done in triplicate from two independent assays, and data were processed using the GraphPad Prism software.

## Plaque assay

A confluent monolayer of HFFs was infected with ~50 freshly egressed parasites for 8 days. Cells were fixed in PBS-3% (vol/vol) formaldehyde as described above and plaques were visualized by Giemsa staining (Sigma-Aldrich; GS500) according to the manufacturer's instructions.

## RNA-seq and sequence alignment

For each biological assay, $3 \times 10^6$ BMDMs were seeded per well in 3 mL DMEM in six-well tissue culture plates. Cells were left uninfected or infected with the Pru *ku80*, Pru *ku80 Δgra18*, and Pru *ku80 Δgra18*, GRA18$^{+++}$ strains (MOI = 6) for 18 hr and subsequently subjected or not to LPS stimulation (100 ng/mL of bacterial lipopolysaccharide from *Escherichia coli* O26:B6; Sigma-Aldrich) for 4 hr.

Total RNAs were extracted and purified using TRIzol (Invitrogen, Carlsbad, CA, USA) and RNeasy Plus Mini Kit (Qiagen). RNA quantity and quality were measured by NanoDrop 2000 (Thermo Scientific). RNA integrity was assessed by standard non-denaturing 1.2% TBE agarose gel electrophoresis. The ribosomal large subunits from *Toxoplasma* and the host cells was used to verify that the ratio between *Toxoplasma* RNA versus host RNA was equivalent between the different biological samples, thus indicating that the samples had equivalent infection rates. For each condition, total RNAs from two independent biological replicates were pooled to prepare cDNA libraries which were then sequenced using Illumina technology in a single replicate dataset.

RNA-sequencing was performed by GENEWIZ (South Plainfield, NJ, USA). Briefly, the RNA quality was checked with an Agilent 2100 Bioanalyzer (Agilent Technologies, Palo Alto, California, USA) and Illumina TruSEQ RNA library prep and sequencing reagents were used following the manufacturer's recommendations (Illumina, San Diego, CA, USA). The samples were paired-end multiplex sequenced (2 × 125 bp) on the Illumina Hiseq 2500 platform and generated at least 70 million reads for each sample (*Supplementary file 3*).

The RNA-Seq reads (FASTQ) were processed and analyzed using the Lasergene Genomics Suite version 14 (DNASTAR, Madison, WI, USA) using default parameters. The paired-end reads were uploaded onto the SeqMan NGen (version 14, DNASTAR. Madison, WI, USA) platform for reference-based assembly using either the *Mus musculus* genome package (GRCm38.p3) or the *Toxoplasma* Type II ME49 strain (ToxoDB-24, ME49 genome) as reference template. The ArrayStar module (version 14, DNASTAR. Madison, WI, USA) was used for normalization, differential gene expression and statistical analysis of uniquely mapped paired-end reads using the default parameters. The expression data quantification and normalization were calculated using the RPKM (Reads Per Kilobase of transcript per Million mapped reads) normalization method. Pathway and functional annotation analyses of the differentially expressed genes were conducted using the hallmark gene sets of GSEA (*Mootha et al., 2003*; *Subramanian et al., 2005*) with default parameters.

## Transfection of RAW264.7 cells

Cells were transfected by electroporation using the Neon Transfection System (Thermo Fisher Scientif). Twenty-four hours before transfection, the RAW264.7 cells were plated at 50% confluency in T-175 tissue culture flask. About 3 × 10^6 cells were washed once in PBS and resuspended in 100 μL of R buffer for transfection. This suspension was mixed with 15 μg of DNA followed by electroporation at 1680 V and 20 ms for one pulse.

## Generation of β-catenin-deficient RAW 264.7 cell line

RAW264.7 cells was transfected with 15 μg of pX330-hSpCas9::sgCTNNB1 vector as described above. At 48 hr after electroporation, cells were cloned by limited dilution and individual clones were expanded and tested for β-catenin expression by IFA and immunoblotting using β-catenin antibodies. The *Ctnnb1* genomic sequences were amplified by PCR using primers Chk ctnnb1_F and Chk ctnnb1_R. The resulting fragments were then ligated into pCR2.1-TOPO for sequencing.

## Chromatographic purification of GRA18-containing complex

Cytoplasmic fraction from T-Rex cells stably expressing Flag-tagged protein or J774 cells infected with Pru ku80 expressing Flag-tagged GRA18 (GRA18+++), were incubated with anti-FLAG M2 affinity gel (Sigma-Aldrich) for 1 hr at 4°C. Beads were washed with 20-column volumes of the BC500 buffer (20 mM Tris-HCl, pH8.0, 0.5 M KCl, 20% Glycerol, 0.5 mM DTT, 0.5% NP40, and protease inhibitors). Bound polypeptides were eluted stepwise with 250 μg/mL FLAG peptide (Sigma Aldrich) diluted in BC100 buffer. For size-exclusion chromatography, protein eluates were loaded onto a Superose 6 HR 10/30 column equilibrated with BC500. Flow rate was fixed at 0.35 mL/min, and 0.5 mL fractions were collected.

## Co-Immunoprecipitation assay

Flag-tagged GRA18 in T-Rex cells was purified by FLAG affinity as follows. Cells were washed in ice-cold PBS and resuspended in buffer A (10 mM Tris-HCl, pH7.9, 1.5 mM MgCl$_2$, 10 mM KCl, 0.5 mM DTT, 0.2 mM PMSF, and 1X Roche protease inhibitor cocktail). Swelling cells were lysed for 8 min with 0.1% NP40 followed by 10 times dounced with loose plunger to disrupt the plasma membranes.

Cell lysates were centrifuged at 1,300 g for 10 min and the supernatant containing the cytosolic fraction was collected.

The cytoplasmic extracts were supplemented with 1:10th volume of 10X buffer B (0.3 M Tris-HCl, pH7.9, 1.4M KCl, 0.03 M MgCl$_2$) before incubation with 50 µL FLAG M2 affinity gel (Sigma-Aldrich) for 1 hr at 4°C. Beads were washed three times in BC500. Bound polypeptides were eluted with 50 µL 250 µg/ml FLAG peptide (Sigma Aldrich) diluted in BC100 buffer. Samples were mixed with protein sample buffer (Invitrogen) for immunoblot analysis.

## Mass spectrometry–based proteomics

Protein bands were excised from colloidal blue–stained gels (Thermo Fisher Scientific), treated with DTT and iodoacetamide to alkylate the cysteines before in-gel digestion using modified trypsin (sequencing grade; Promega). Resulting peptides from individual bands were analyzed by nanoLC-MS/MS (UltiMate 3000 coupled to LTQ-Orbitrap Velos Pro; Thermo Fisher Scientific) using a 25 min gradient. Peptides and proteins were identified and quantified using MaxQuant (version 1.5.3.17) through concomitant searches against ToxoDB (20151112 version), SwissProt (Homo sapiens taxonomy, 20151112 version), and the frequently observed contaminant database embedded in MaxQuant. Minimum peptide length was set to seven amino acids. Minimum number of peptides, razor +unique peptides, and unique peptides were all set to 1. Maximum false discovery rates were set to 0.01 at peptide and protein levels.

## Immunoblot analysis

Immunoblot analysis of protein was performed as described in *Bougdour et al. (2013)*. Briefly, ~10$^7$ cells were lysed in 50 µL lysis buffer (10 mM Tris-HCl, pH6.8, 0.5% SDS [v/v], 10% glycerol [v/v], 1 mM EDTA and protease inhibitors cocktail) and sonicated. The protein extracts were separated by SDS-PAGE, and transferred to a PolyVinyliDene Fluoride membrane (PVDF; immobilon-P, Millipore) by liquid transfer. M were then blocked with PBS buffer containing 0.01% Tween 20 (v/v) and 5% nonfat dry milk. Appropriate primary antibodies diluted in PBS containing 0.03% Tween 20 (v/v) were used to probe the membrane. Primary antibodies were detected using alkaline phosphatase or horseradish peroxidase conjugated secondary antibodies with Novex AP Chromogenic Substrate (Invitrogen) or Supersignal West Pico Chemiluminescent Substrate kit (Thermo Fisher Scientific), respectively.

## Quantitative real-time PCR

Total RNA was isolated using TRIzol reagent (Thermo Fisher Scientific) and RNA quality and integrity was assessed by agarose gel electrophoresis and quantification of 28S and 18S ribosomal RNA. cDNA was synthetized using the High Capacity RNA-to-cDNA kit (Applied Biosystem). Samples were analyzed by real time quantitative PCR for *Ccl17*, *Ccl22*, *Ccl24* and *Ifnb1* using TaqMan Gene Expression Master Mix (Applied Biosystems) according to the manufacturer's instructions. *Tbp* was used as an internal control gene for normalization. Data were analyzed using the 2$^{-\Delta\Delta Ct}$ method, and statistical analyses were conducted on the ΔCt values. Data are represented as mean value ± s.d. of three technical replicates and the results presented are representative of at least two independent experiments.

## In vitro cytokine ELISA

About 10$^6$ BMDMs were seeded per well of a 24-well-plate. Cells were left uninfected or infected with freshly egressed *T. gondii* tachyzoites at an MOI of 1:5. At 24 hr post-infection, supernatants were collected and stored at −80°C for storage. CCL17 and CCL22 levels were determined using Mouse CCL17/TARC and Mouse CCL22/MDC Quantikine ELISA Kits from R and D Systems according to the manufacturer's instructions. Data are mean value ± s.d. of four independent experiments from two biological replicates.

## Acknowledgments

This work was supported by the ANR Jeune Chercheur 2012 ToxoEffect (ANR-12-JSV3–0004 – 01), the European Research Council (ERC Consolidator Grant No 614880 Hosting TOXO), and the LabEx ParaFrap (ANR-11-LABX-0024). The authors have no competing financial interests.

## Additional information

### Funding

| Funder | Grant reference number | Author |
| --- | --- | --- |
| European Commission | ERC Consolidator Grant No. 614880 HostingToxo | Laurence Braun<br>Mohamed-Ali Hakimi<br>Huan He |
| Agence Nationale de la Recherche | LabEx ParaFrap (ANR-11-LABX-0024) | Mohamed-Ali Hakimi<br>Alexandre Bougdour<br>Marie-Pierre Brenier-Pinchart<br>Laurence Braun |
| Agence Nationale de la Recherche | ANR-12-JSV3-0004-01 | Alexandre Bougdour |

The funders had no role in study design, data collection and interpretation, or the decision to submit the work for publication.

### Author contributions

Huan He, Data curation, Formal analysis, Validation, Investigation, Visualization; Marie-Pierre Brenier-Pinchart, Formal analysis, Supervision, Validation, Investigation, Visualization; Laurence Braun, Formal analysis, Validation, Investigation, Visualization; Alexandra Kraut, Data curation, Formal analysis, Validation, Investigation; Bastien Touquet, Formal analysis, Investigation, Visualization, BT performed the experiments conducted with the ScanR microscope system; Yohann Couté, Supervision, Validation, Investigation; Isabelle Tardieux, Supervision, Writing—original draft, Writing—review and editing; Mohamed-Ali Hakimi, Conceptualization, Resources, Formal analysis, Funding acquisition, Writing—original draft, Project administration; Alexandre Bougdour, Conceptualization, Resources, Data curation, Formal analysis, Supervision, Funding acquisition, Validation, Investigation, Visualization, Writing—original draft, Project administration, Writing—review and editing

### Author ORCIDs

Yohann Couté http://orcid.org/0000-0003-3896-6196
Isabelle Tardieux http://orcid.org/0000-0002-5677-7463
Mohamed-Ali Hakimi http://orcid.org/0000-0002-2547-8233
Alexandre Bougdour http://orcid.org/0000-0002-5895-0020

### Ethics

Animal experimentation: This study was performed under pathogen-free conditions in accordance with established institutional guidance and approved protocols from the institutional animal care and use committee protocol (#175_UHTA-UMR5163-AB-01) of the University Grenoble Alpes. For all the experiments performed, every effort was made to minimize suffering.

### Decision letter and Author response

Decision letter https://doi.org/10.7554/eLife.39887.026
Author response https://doi.org/10.7554/eLife.39887.027

## Additional files

### Supplementary files

• Supplementary file 1. Strains and Plasmids, Primers and oligonucleotides.

DOI: https://doi.org/10.7554/eLife.39887.013

• Supplementary file 2. Y2H screen results.

DOI: https://doi.org/10.7554/eLife.39887.014

• Supplementary file 3. Gene expression profiles in BMDMs and *T. gondii* with and without LPS stimulation. RNA-Seq Reads Mouse. Summary of total RNASeq reads and average reads mapped to the mouse genome. RNA-Seq Reads *Toxoplasma*. Summary of total RNASeq reads and average reads mapped to the *Toxoplasma* genome. RPKM values for BMDMs. Expression values for all the mouse genes in the indicated samples. RPKM and log2 transformed values are shown. RPKM values for *T. gondii*. Expression values for all the *Toxoplasma* genes in the indicated samples. RPKM and log2 transformed values are shown. Pru vs *gra18*. RPKM values of the BMDM genes differentially regulated between the wild-type (Pru *ku80*) and *gra18* mutant strains. Genes that were modulated with more than three-fold change and having a signal threshold above 5 RPKM in at least one sample when comparing the wild-type and Δ*gra18* mutant strains are shown. RPKM and log2 transformed values are shown for the indicated samples. ui vs ui_LPS. RPKM values of the BMDM genes differentially regulated between uninfected BMDMs that were left unstimulated (ui) or stimulated with LPS for 4 hr (ui_LPS). Genes that were modulated with more than three-fold change and having a signal threshold above 5 RPKM in at least one sample when comparing the ui and ui_LPS samples are shown. RPKM and log2 transformed values are shown for the indicated samples. β-Catenin target genes. RPKM values of known β-catenin/TCF target genes in the RNA-Seq experiment presented in *Figure 5*. Some direct target genes defined as those with Tcf binding sites are indicated in red. ui vs Pru. RPKM values of the BMDM genes differentially regulated when comparing BMDM left uninfected (ui) and BMDM infected by wild-type parasites (Pru) in the absence of LPS stimulation. Genes that were modulated with more than three-fold change and having a signal threshold above 5 RPKM in at least one sample when comparing the ui and Pru samples are shown. RPKM and log2 transformed values are shown for the indicated samples.

DOI: https://doi.org/10.7554/eLife.39887.015

• Transparent reporting form

DOI: https://doi.org/10.7554/eLife.39887.016

## Data availability

Datasets Generated: Transcriptomic analysis by Next Generation Sequencing (RNA-seq) have been deposited in GEO under accession code GSE103113.

The following dataset was generated:

| Author(s) | Year | Dataset title | Dataset URL | Database and Identifier |
|---|---|---|---|---|
| He H, Brenier-Pinchart M, Braun L, Kraut A, Touquet B, Couté Y, Tardieux I, Hakimi M, Bougdour A | 2018 | Transcriptomic analysis by Next Generation Sequencing of mouse bone marrow derived macrophages (BMDMs) infected by Wild-Type and gra18 mutant strains of T. gondii. | https://www.ncbi.nlm.nih.gov/geo/query/acc.cgi?acc=GSE103113 | NCBI Gene Expression Omnibus, GSE103113 |

The following previously published datasets were used:

| Author(s) | Year | Dataset title | Dataset URL | Database and Identifier |
|---|---|---|---|---|
| Melo MB, Nguyen QP, Cordeiro C, Hassan MA, Yang N, McKell R, Rosowski EE, Julien L, Butty V, Darde M, Ajzenberg D, Fitzgerald K, Young LH, Saeij JPJ | 2013 | Toxoplasma gondii Transcriptome or Gene expression | https://trace.ncbi.nlm.nih.gov/Traces/sra/?study=SRP011061 | NCBI Sequence Read Archive, SRP011061 |
| Hammoudi P, Jacot D, Mueller C, Cristina MD, Dogga S, | 2015 | RNA-Seq analysis of mouse BMDMs infected RH and Pru strain Asp5-KO parasites | https://www.ebi.ac.uk/ena/data/view/PRJEB10909 | ENA - European Nucleotide Archive, PRJEB10909 |

Marq J, Romano J, Tosetti N, Dubrot J, Emre Y, Lunghi M, Coppens I, Yamamoto M, Sojka D, Pino P, Soldati-Favre D

| | | | | |
|---|---|---|---|---|
| Melo MB, Nguyen QP, Cordeiro C, Hassan MA, Yang N, McKell R, Rosowski EE, Julien L, Butty V, Darde M, Ajzenberg D, Fitzgerald K, Young LH, Saeij JPJ | 2013 | Whole-genome sequencing describes recombination in Toxoplasma and identifies loci that determine fitness and avoidance of outcrossing | https://trace.ncbi.nlm.nih.gov/Traces/sra/?study=SRP008923 | NCBI Sequence Read Archive, SRP008923 |

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
