## [Decision Letter]

[Editors’ note: a previous version of this study was rejected after peer review, but the authors submitted for reconsideration. The first decision letter after peer review is shown below.]

Thank you for submitting your work entitled "Characterization of a *Toxoplasma*effector uncovers an alternative β-catenin-regulatory pathway of inflammation" for consideration by *eLife*. Your article has been reviewed by a Senior Editor, a Reviewing Editor and three peer reviewers. The following individuals involved in review of your submission have agreed to reveal their identity: Jon P Boyle (Reviewer #1).

Our decision has been reached after consultation between the reviewers. Based on these discussions and the individual reviews below, we regret to inform you that your work will not be considered at this stage for publication in *eLife*.

We all agreed that the reported observations are potentially interesting however some serious concerns were raised including notably the artificial nature of the experimental conditions that always depend on ectopic overexpression of GRA18. Without verifying the effects when expressed at endogenous levels or via KO, the biological relevance of the findings is not clear.

While we are rejecting the paper as a result of the criticisms, if you feel that you are able to address these issues, we encourage you for a resubmission that will be handled as revision.

Reviewer #1:

Subsection “GRA18 is secreted and exported to the cytoplasm of infected host cells”: Should be more specific about classes of secreted effectors that do or do not go to the host cell nucleus.

Subsection “GRA18 is secreted and exported to the cytoplasm of infected host cells”: "identity" not "identities" (similar grammatical errors throughout) a couple of places GRA18 is referred to as GRA16.

This is a very thorough and exciting manuscript identifying a new effector secreted by *Toxoplasma gondii* as well as a new target pathway that, in murine cells, induces the production of regulatory chemokines like CCL17, CCL22, and CCL24. While many secreted dense granule effectors traffic to the host cell nucleus, GRA18 remains in the host cell cytoplasm, where it interacts with components of the β-catenin signaling pathway, leading to activation and translocation of β-catenin into the host cell nucleus. GRA18 itself interacts with these components both during infections and in ectopic expression, indicating that it is required for manipulation of this pathway. overall the work is of high quality and experiments are well-controlled and interpreted properly.

While it is clear that gra18 interacts with β-catenin and β-catenin signaling components in human and murine cells, the gra18-driven chemokine transcription and/or chemokine secretion was not determined in human cells. This would either establish the chemokine effect as being broadly applicable to cells from multiple species or would suggest that distinct effectors may be responsible for chemokine induction in cells from different species. There is certainly precedent for mouse and human responses to certain effectors being distinct (IRGs vs. rhoptry proteins; profilin vs. Toll-like receptors, e.g.).

It would be helpful if the in vivo mouse infection data comparing gra18 wt/ko would be shown or described in more detail (as written it is discussed as being slightly less virulent but this should be quantified).

Are there statistical analyses in Figure 6? while the data showing that β-catenin KO cells don't make as much *Ccl17* or *Ccl24*, the data are less clear for *Ccl22*.

For the RNA-Seq experiments I cannot tell how many replicates were performed for each condition. this should be included in the methods and it would be helpful if the heat map in Figure 6 could include all replicates (are they averages? or was there only a single RNA-Seq replicate?)

Reviewer #2:

The mechanisms by which intracellular microbes hijack the host cell is a fascinating area of study. Here the authors explore a novel effector protein produced by the protozoan parasite *Toxoplasma*. The authors find that this effector, GRA18, can form complexes with the Wnt effector β-catenin and its regulators GSK3 and PP2A. They find that overexpressing GRA18 can stabilize β-catenin and lead to downstream effects on genes expression. The results are generally solid, but, as outlined below, I felt there was one major conceptual flaw that undercut the study and reduced the insights available-this significantly reduced my enthusiasm. I also outline below some other experimental concerns.

The authors present very solid data demonstrating that GRA18 can bind to and coIP with β-catenin, GSK3 and PP2A. They also show that overexpression of GRA18 can stabilize β-catenin and lead to downstream effects on genes expression. However, as they admit, loss of GRA18 has no detectable effects on β-catenin levels, gene expression or parasite proliferation. This may be due to redundancy with other effectors but as a result there is no evidence that role they suggest is a natural one. Anything that tightly bound and sequestered GSK3 would be likely to increase β-catenin levels when overexpressed, even if this was not its normal function. This issue substantially undercut the biological relevance of these findings and meant that many of the authors strong conclusions should be phrased as what GRA18 "can do" rather than what it "does".

Figure 1C. The subcellular localization of GRA18 expressed from two different promotors was very different. This was troubling.

Figure 3. The binding of GRA18 to β-catenin as assessed by 2 hybrid and by coIP gave different results.

Reviewer #3:

This is an interesting report that describes the effects of a secreted parasite protein on host cell signaling. The effector described here (GRA18) is novel as is implication of the β-catenin pathway that it proposed to modulate. I feel that the work is carefully done but relies too heavily on over-expression and heterologous expression and hence I have some concerns about the validity of the conclusions.

The lack of a striking phenotype in infected mice for gra18 mutants suggest that the modulation of the β-catenin pathway described here does not play a major role in pathogenesis. This lack of an overt phenotype should encourage the authors to consider other roles for the effector they have defined. For example, given the chemokine signals evoked here, are there differences in cellular recruitment linked to GRA18?

One concern about the interactions studied here is that they may be driven by mass action as many of the key experiments are performed by over-expressing the effector either for Y2H or using the inducible T-Rex system. Even the confirmation studies using host cells infected with *T. gondii* appear to use a construct that is over-expressed (subsection “GRA18 forms versatile complexes with host components of the β-catenin destruction 112 complex”). Therefore, there is some concern that these interactions are not normally triggered during infection. Paralleling this pattern, it was only when GRA18 was strongly over-expressed (subsection “GRA18 functions as a positive regulator of β-catenin”) that an effect on β-catenin could be shown in infected cells.

Although it seems clear from RNA-Seq studies that the presence of GRA18 drives induction of chemokines such as *Ccr7, Ccl1, Ccl17* and *Ccl22*, – these are not typical Wnt-β-catenin-dependent genes. Thus, it would be important to show that their induction by GRA18 (or loss in gra18 mutants) is actually β-catenin-dependent. Again, this experiment only seems to have been performed with ectopic over-expression.

I feel that if the pathway describe here is robust the authors should be able to do many of these experiments with infection of WT and mutant strains and examine gene induction by qRT-PCR or cytokine levels by luminex of similar technology. Absent this, there is some chance that the apparent modulation of β-catenin is only due to over-expression and the true role of GRA18 is chemokine modulation through another pathway.

[Editors’ note: what now follows is the decision letter after the authors submitted for further consideration.]

Thank you for submitting your article "Characterization of a *Toxoplasma* effector uncovers an alternative GSK3/β-catenin-regulatory pathway of inflammation" for consideration by *eLife*. Your article has been reviewed by Gisela Storz as the Senior Editor, a Reviewing Editor and three peer reviewers. The reviewers have opted to remain anonymous.

The reviewers have discussed the reviews with one another and the Reviewing Editor has drafted this decision to help you prepare a revised submission.

Summary:

The authors report here an extensive characterization of a new dense granule protein from *Toxoplasma gondii*, GRA18, that traffics to the host cell cytosol where, remarkably, it binds to several proteins in the β-catenin destruction complex, including β-catenin. GRA18 affects β-catenin signaling via increased β-catenin activity and expression of many host genes including specific chemokines. The results are impressive in their scope and the multiple ways in which most of the conclusions are supported by independent experiments. Overall the model is supported by the data presented and the paper is clearly and rigorously written, especially given the complexity of the story being presented. It ends with a mystery in terms of why the genes that are classically considered to be downstream of β-catenin are not activated by GRA18's action on β-catenin. The authors are forced to invoke another, as yet unidentified, parasite protein as playing a role that intersects with GRA18 in an infected cell. Although the exact mechanism of interaction is not yet understood, the data shown is solid and in general support the interpretation of GRA18 as the driving force of a non-canonical β-catenin-dependent signaling pathway conserved over multiple host species.

Essential revisions:

While a great deal of data is provided, some of the conclusions reached do not always appear to be strongly supported by the data.

1). GRA18 binding- The authors state that GRA18 directly binds β-catenin though I think the data they provide is certainly less robust than the data they have for GSK and PPP2R5A/B (e.g. the original Y2H found 3 clones showing β-catenin interactions with "B" ratings as opposed to GSK and PPP2R5A/B- which had over 10 clones with "A" ratings; the Y2H with bits of GRA18 shows a direct β-catenin interaction but the infected macrophage data and T-rex with overexpression do not). The authors need to either provide stronger explanations/data about the direct binding to β-catenin or soften their conclusion (and possibly to change the model- Figure 8).

2) Subsection “GRA18 functions as a positive regulator of β-catenin” – The virulence data shown is from one experiment and there is no mention in the legend or text if this was repeated. Given the variability that can be seen in animal experiments, at least one repeat is needed.

3) Chemokine data- Between Figure 5, Figure 6, and Figure 7 there are inconsistencies. Figure 5E and F show that *Ccl17* and *Ccl22* have a GRA18-dependent increase in expression and protein levels but *Ccl24* only changes at the level of transcript. Instead of exclusively focusing on *Ccl17* and *Ccl22*, all 3 are noted in Figure 6 (using only expression data) with Figure 6F suggesting that *Ccl22* changes in GRA18-β-catenin-independent mechanism. This idea seems to be contradicted in Figure 7 where the authors use *Ccl22* as a readout for GRA18 inhibiting GSK3 which leads to increased β-catenin activity which then drives over-expression of *Ccl22*. These discrepancies need to be addressed including in the model.

4) Subsection “Quantitative real-time PCR”: Was the RNA quality/integrity controlled and *Tbp* as the only control gene tested for its expression stability? Not doing so could potentially undermine the robustness and reliability of the qPCR data. Why is only 1 biological replicate shown in the qPCR data and not a combination of 2 or more?

---

## [Author Response]

[Editors’ note: the author responses to the first round of peer review follow.]

Our decision has been reached after consultation between the reviewers. Based on these discussions and the individual reviews below, we regret to inform you that your work will not be considered at this stage for publication in eLife.We all agreed that the reported observations are potentially interesting however some serious concerns were raised including notably the artificial nature of the experimental conditions that always depend on ectopic overexpression of GRA18. Without verifying the effects when expressed at endogenous levels or via KO, the biological relevance of the findings is not clear.

We fully concur with the reviewers that our statement about GRA18 functional dependency on β-catenin relies heavily on overexpression systems that could lead to an artificial accumulation of β-catenin, otherwise not affected upon infection by either wild-type or *gra18* mutant parasites. To clarify this issue, first, GRA18 protein complex was conducted using endogenously tagged GRA18 instead of a GRA18 overexpressor strain to validate the interactions with the β-catenin destruction complex in a more natural context. Interaction with both GSK3 and PP2A-B56 were confirmed, but the β-catenin was not detected in the eluate, as expected, given the loosen interaction observed with this substrate by FPLC analysis. In a second step, chemokine expression was assayed in RAW cells mutated or not for *Ctnnb1* as suggested by the reviewers. The results are now included in Figure 6.

**Author response image 1. respfig1:** qRT-PCR analysis of MAF-DKO macrophages (low passages) infected by the indicated strains of *T. gondii* at a MOI of 1:6.

While the results obtained for *Ccl17* mirror those obtained by ectopic overexpression, differences were observed for *Ccl22*. In contrast to BMDM cells setting, the induction of *Ccl22* by *T. gondii* is only partially dependent on GRA18, and as expected poorly- if not at all- dependent on β-catenin. Given the discrepancy in the magnitude of the expression of the chemokine when comparing RAW with BMDM cells, we searched for another murine macrophage more closely related to the BMDMs. We obtained the MAF-DKO cells derived from BMDMs (Aziz et al., 2009) from Michael Sieweke’s lab, and checked that they responded quite well to *T. gondii* infection with regard to chemokine induction (see Author response image 1). Therefore, we undertook the construction of a *Ctnnb1* mutant in these cells using lentiviral systems (First lentivirus for the CAS9 and a second for introducing the specific gRNA). After a selection process over two months of the different lentiviral constructions (CAS9-Blast. and gRNA-Puro.), we observed that the cells expressing the CAS9 only (MAK-DKO wt) expressed *Ccl17* and *Ccl22* only at low levels upon infection. Only *Ccl24* gave relevant results (Author response image 2). Given the possibility that the cells have derived over passages, we decided not to include these results in the manuscript. We dampened our conclusion about the function played by β-catenin in the regulation of these chemokines and emphasized the role of the GRA18/β-catenin pathway in the regulation of *Ccl17* and the discrepancy with the regulation of *Ccl22*. We also clarified in the text the GRA18independent regulation of *Ccl22* subsection “GRA18 alters host gene expression in a β-catenin-dependent fashion” and subsection “The GRA18-GSK3-β-catenin axis induces the expression of anti-inflammatory chemokines” (see response to the reviewer #1). Please note that a specific response has also been provided below in response to reviewer #2.

**Author response image 2. respfig2:** qRT-PCR analysis of MAF-DKO macrophages (high passages, after lentiviral transductions) infected by the indicated strains of *T. gondii* at a MOI of 1:6.

While we are rejecting the paper as a result of the criticisms, if you feel that you are able to address these issues, we encourage you for a resubmission that will be handled as revision.Reviewer #1:Subsection “GRA18 is secreted and exported to the cytoplasm of infected host cells”: Should be more specific about classes of secreted effectors that do or do not go to the host cell nucleus.

An exhaustive list of effectors targeted to host cell nucleus was added to subsection “GRA18 is secreted and exported to the cytoplasm of infected host cells”.

Subsection “GRA18 is secreted and exported to the cytoplasm of infected host cells”: "identity" not "identities" (similar grammatical errors throughout) a couple of places GRA18 is referred to as GRA16.

Corrected.

This is a very thorough and exciting manuscript identifying a new effector secreted by Toxoplasma gondii as well as a new target pathway that, in murine cells, induces the production of regulatory chemokines like CCL17, CCL22, and CCL24. While many secreted dense granule effectors traffic to the host cell nucleus, GRA18 remains in the host cell cytoplasm, where it interacts with components of the β-catenin signaling pathway, leading to activation and translocation of β-catenin into the host cell nucleus. GRA18 itself interacts with these components both during infections and in ectopic expression, indicating that it is required for manipulation of this pathway. overall the work is of high quality and experiments are well-controlled and interpreted properly.While it is clear that gra18 interacts with β-catenin and β-catenin signaling components in human and murine cells, the gra18-driven chemokine transcription and/or chemokine secretion was not determined in human cells. This would either establish the chemokine effect as being broadly applicable to cells from multiple species or would suggest that distinct effectors may be responsible for chemokine induction in cells from different species. There is certainly precedent for mouse and human responses to certain effectors being distinct (IRGs vs. rhoptry proteins; profilin vs. Toll-like receptors, e.g.).

We agree that host specificity is an interesting issue and haven’t yet looked much to human cell types other than HFF to assay chemokine expression. As recently published by Jon Boyle’s group (Ander et al., 2018), the chemokines *Ccl17* and *Ccl22* were not found induced upon *T. gondii* infection of HFFs cells. However, given the induction of CCL17 and CCL22 in human placental cells, one might wonder whether this is a general function of GRA18 and/or restricted to certain cell types. We rephrased and expanded the discussion on this point (see subsection “The GRA18-GSK3-β-catenin axis induces the expression of anti-inflammatory chemokines”).

It would be helpful if the in vivo mouse infection data comparing gra18 wt/ko would be shown or described in more detail (as written it is discussed as being slightly less virulent but this should be quantified).

As suggested by the reviewer, the in vivo mouse infection data were added to the Figure 4.

Are there statistical analyses in Figure 6? while the data showing that β-catenin KO cells don't make as much Ccl17 or Ccl24, the data are less clear for Ccl22.

Statistics were added to the Figure 6.

For the RNA-Seq experiments I cannot tell how many replicates were performed for each condition. this should be included in the methods and it would be helpful if the heat map in Figure 6 could include all replicates (are they averages? or was there only a single RNA-Seq replicate?)

To clarify this important technical issue, please note that each condition was assayed by single RNA-Seq replicate of RNA samples that were obtained from two independent experiments that were pooled before library construction. In our data set, two conditions were experienced, with or without LPS stimulation. The impact of GRA18 on host cell gene expression does not interfere with the LPS response (Supplementary file 3) and therefore the expression level of the genes regulated by GRA18 remained similar when comparing unstimulated or LPS treated samples, which are both shown in the Figure 5A (compare unstimulated Pru *ku80* with LSP treated Pru *ku80*, etc.). As a result, the genes that were found differentially regulated when comparing WT and *gra18* mutant parasites in the absence of LPS, were also found differentially regulated in the presence of LPS. We certainly agree that the number of biological replicates is a limitation, but we have been careful to perform independent assays using RNA-Seq comparative approach and different conditions (i.e. with and without LPS). The data were systematically verified not only by RT-qPCR assays from different biological replicates shown in Figure 5E, but also by ELISA in Figure 5F. Therefore, we feel confident in the reproducibility of our results.

Reviewer #2:The mechanisms by which intracellular microbes hijack the host cell is a fascinating area of study. Here the authors explore a novel effector protein produced by the protozoan parasite Toxoplasma. The authors find that this effector, GRA18, can form complexes with the Wnt effector β-catenin and its regulators GSK3 and PP2A. They find that overexpressing GRA18 can stabilize β-catenin and lead to downstream effects on genes expression. The results are generally solid, but, as outlined below, I felt there was one major conceptual flaw that undercut the study and reduced the insights available-this significantly reduced my enthusiasm. I also outline below some other experimental concerns.The authors present very solid data demonstrating that GRA18 can bind to and coIP with β-catenin, GSK3 and PP2A. They also show that overexpression of GRA18 can stabilize β-catenin and lead to downstream effects on genes expression. However, as they admit, loss of GRA18 has no detectable effects on β-catenin levels, gene expression or parasite proliferation. This may be due to redundancy with other effectors but as a result there is no evidence that role they suggest is a natural one. Anything that tightly bound and sequestered GSK3 would be likely to increase β-catenin levels when overexpressed, even if this was not its normal function. This issue substantially undercut the biological relevance of these findings and meant that many of the authors strong conclusions should be phrased as what GRA18 "can do" rather than what it "does".

While we agree that most of the demonstration about GRA18 function was performed using GRA18 overexpression systems (see the above response), it is important to consider that the downstream effects on gene expression do not necessarily depend on GRA18 overexpression as they were originally found by comparing WT and gra18 mutant parasites in Figure 5. To strengthen our data, several new sets of data have been added:

1) We present now evidence that GRA18 forms a complex with GSK3 and PP2A-B56 subunit upon infection with parasites expressing GRA18 fusion protein encoded from the endogenous locus (new panel in Figure 2).

2) We present evidence that even though β-catenin is not up-regulated upon infection by WT parasites, β-catenin basal levels are yet necessary for GRA18 functionality towards *Ccl17* when comparing macrophages infected by WT or gra18 mutant parasites. While dependency on β-catenin is clear for *Ccl17*, the situation is different for *Ccl22*, thus suggesting different underlying mechanisms.

In the view of these new datasets, the conclusion was rephrased throughout the manuscript (subsection “GRA18 forms versatile complexes with host components of the β-catenin destruction complex”, subsection “GRA18 alters host gene expression in a β-catenin-dependent fashion”, subsection “GRA18 export mechanism”, and subsection “The GRA18-GSK3-β-catenin axis induces the expression of anti-inflammatory chemokines”), and the model presented in Figure 8 has been updated to integrate the possibility of other putative *T. gondii* factor(s) to be involved in chemokine regulation.

Figure 1C. The subcellular localization of GRA18 expressed from two different promotors was very different. This was troubling.

The expression of GRA18 is much weaker than the GRA16 and TgIST effectors. To exacerbate the detection of GRA18 in the cytoplasm of the infected host cells, we chose to use the strong and widely used GRA1 promoter to drive GRA18 expression in the parasites, which clearly leads to GRA18 overexpression at both the transcriptional (Figure 5D) and at the protein levels (Figure 1C). A direct consequence is an accumulation of GRA18 in the parasite cytoplasm and the PV space presumably due to overloading the export process across the PVM. To accurately visualize the accumulation of GRA18 within the PV space, cells were permeabilized with ethanol instead of Triton X-100 (used for GRA18HA3, upper panel), which allowed better visualization of the PV space between the PVM and the PV enclosed zoites. Although there are some differences in the GRA18 staining pattern, the Figure 1C clearly shows the presence of GRA18 in the host cytoplasm, which was the primary aim of this assay. The text of the result section (subsection “GRA18 is secreted and exported to the cytoplasm of infected host cells”) and Figure legend were modified accordingly.

Figure 3. The binding of GRA18 to β-catenin as assessed by 2 hybrid and by coIP gave different results.

We find this somewhat puzzling as well, but basically, the two assays use different fragment lengths (~200 aa for the Y2H and >400 aa for coIP) and were performed under different conditions; one is in vivo and the other an in vitro assay using much stringent conditions when considering the amount potassium chloride present (0.5 M KCl for the coIP assay). While the 2YH assay can detect interactions between N-ter fragment of GRA18 with β-catenin, the coIP did not. The coIP pulled down GSK3 with the C-ter domain of GRA18 while the small fragments used in the Y2H did not. Although these results differ, they are not contradictory or mutually exclusive and should be analyzed together to extract as much information as possible. Given that both assays agree with the binding of the PP2A subunit to the C-ter domain of GRA18, these results indicate that B56 binds to the extreme C-ter part of GRA18 which is fully consistent with the presence of putative B56 binding sites in this region (LxxIxE motifs).

Reviewer #3:This is an interesting report that describes the effects of a secreted parasite protein on host cell signaling. The effector described here (GRA18) is novel as is implication of the β-catenin pathway that it proposed to modulate. I feel that the work is carefully done but relies too heavily on over-expression and heterologous expression and hence I have some concerns about the validity of the conclusions.The lack of a striking phenotype in infected mice for gra18 mutants suggest that the modulation of the β-catenin pathway described here does not play a major role in pathogenesis. This lack of an overt phenotype should encourage the authors to consider other roles for the effector they have defined. For example, given the chemokine signals evoked here, are there differences in cellular recruitment linked to GRA18?One concern about the interactions studied here is that they may be driven by mass action as many of the key experiments are performed by over-expressing the effector either for Y2H or using the inducible T-Rex system. Even the confirmation studies using host cells infected with T. gondii appear to use a construct that is over-expressed (subsection “GRA18 forms versatile complexes with host components of the β-catenin destruction 112 complex”). Therefore, there is some concern that these interactions are not normally triggered during infection. Paralleling this pattern, it was only when GRA18 was strongly over-expressed (subsection “GRA18 functions as a positive regulator of β-catenin”) that an effect on β-catenin could be shown in infected cells.Although it seems clear from RNA-Seq studies that the presence of GRA18 drives induction of chemokines such as Ccr7, Ccl1, Ccl17 and Ccl22, – these are not typical Wnt-β-catenin-dependent genes. Thus, it would be important to show that their induction by GRA18 (or loss in gra18 mutants) is actually β-catenin-dependent. Again, this experiment only seems to have been performed with ectopic over-expression.I feel that if the pathway describe here is robust the authors should be able to do many of these experiments with infection of WT and mutant strains and examine gene induction by qRT-PCR or cytokine levels by luminex of similar technology. Absent this, there is some chance that the apparent modulation of β-catenin is only due to over-expression and the true role of GRA18 is chemokine modulation through another pathway.

See responses above.

[Editors' note: the author responses to the re-review follow.]

Summary:The authors report here an extensive characterization of a new dense granule protein from Toxoplasma gondii, GRA18, that traffics to the host cell cytosol where, remarkably, it binds to several proteins in the β-catenin destruction complex, including β-catenin. GRA18 affects β-catenin signaling via increased β-catenin activity and expression of many host genes including specific chemokines. The results are impressive in their scope and the multiple ways in which most of the conclusions are supported by independent experiments. Overall the model is supported by the data presented and the paper is clearly and rigorously written, especially given the complexity of the story being presented. It ends with a mystery in terms of why the genes that are classically considered to be downstream of β-catenin are not activated by GRA18's action on β-catenin. The authors are forced to invoke another, as yet unidentified, parasite protein as playing a role that intersects with GRA18 in an infected cell. Although the exact mechanism of interaction is not yet understood, the data shown is solid and in general support the interpretation of GRA18 as the driving force of a non-canonical β-catenin-dependent signaling pathway conserved over multiple host species.Essential revisions:While a great deal of data is provided, some of the conclusions reached do not always appear to be strongly supported by the data.1) GRA18 binding- The authors state that GRA18 directly binds β-catenin though I think the data they provide is certainly less robust than the data they have for GSK and PPP2R5A/B (e.g. the original Y2H found 3 clones showing β-catenin interactions with "B" ratings as opposed to GSK and PPP2R5A/B- which had over 10 clones with "A" ratings; the Y2H with bits of GRA18 shows a direct β-catenin interaction but the infected macrophage data and T-rex with overexpression do not). The authors need to either provide stronger explanations/data about the direct binding to β-catenin or soften their conclusion (and possibly to change the model- Figure 8).

While we certainly agree with the reviewer that the data gathered with biochemical approaches support a relatively weaker interaction of GRA18 with β-catenin unlike with GSK3 or the PP2A regulatory subunit, we want to stress important findings that strongly argue for GRA18/β-catenin interaction. While this interaction was initially found by Y2H, it was indeed subsequently confirmed with two other assays: (i) in HFF cells infected by parasites overexpressing GRA18-HF (Figure 2B, 8 peptides) and (ii) in T-Rex cells ectopically expressing GRA18, therefore unambiguously establishing the partnership (Figure 2E). It is only when GRA18 was expressed under the control of its endogenous promoter that we could detect GRA18 binding to GSK3 and PP2A, but not to β-catenin. Our interpretation of these results is that overexpression of GRA18 led to substantial β-catenin stabilization and accumulation, thus rendering possible its detection by immunoprecipitation otherwise below the limit of detection. Note that this point was mentioned throughout the second version of the manuscript, subsection “GRA18 forms versatile complexes with host components of the β-catenin destruction

complex.” and subsection “GRA18 export mechanism”. To take into account the reviewer’s concerns, we have modified the model proposed in Figure 8 and the corresponding legend to make clear that β-catenin while strongly binding to GRA18 – this interaction resisting to stringent washing conditions (0.5 M KCl and 0.1% NP-40)- only binds in a dose-dependent and sub-stoichiometrical fashion in the context of infection.2) Subsection “GRA18 functions as a positive regulator of β-catenin” – The virulence data shown is from one experiment and there is no mention in the legend or text if this was repeated. Given the variability that can be seen in animal experiments, at least one repeat is needed.

The virulence data in mice presented in Figure 4D are actually the cumulative results of two independent experiments, each conducted at several month intervals. Therefore, the groups of mice under analysis as well as the parasite handling were independent. We now mention these important precisions in the figure legend. We totally agree with the precautions required prior to conclude when assays rely on animal, in particular taking into account the variability between assays. It is indeed why we chose to assay (and to confirm) our findings using two different type II background – i.e. Pru *ku80* and 76K strains – in which we have produced mutants of interest.

3) Chemokine data- Between Figure 5, Figure 6, and Figure 7 there are inconsistencies. Figure 5E and F show that Ccl17 and Ccl22 have a GRA18-dependent increase in expression and protein levels but Ccl24 only changes at the level of transcript. Instead of exclusively focusing on Ccl17 and Ccl22, all 3 are noted in Figure 6 (using only expression data) with Figure 6F suggesting that Ccl22 changes in GRA18-β-catenin-independent mechanism. This idea seems to be contradicted in Figure 7 where the authors use Ccl22 as a readout for GRA18 inhibiting GSK3 which leads to increased β-catenin activity which then drives over-expression of Ccl22. These discrepancies need to be addressed including in the model.

The contribution of GRA18 in regulating *Ccl17, Ccl22*, and *Ccl24* chemokines in BMDMs is indisputable, but the data do not authorize a clear-cut conclusion about the actual reliance on β-catenin for the regulation of *Ccl22*, particularly. The use of BMDMs mutant for *Ctnnb1* and direct transfection of our different GRA18 constructions (Nt vs. Ct) and infection with wild-type and *gra18* mutant parasites would have provided a straightforward demonstration of the contribution of GRA18 and β-catenin, but construction of such a mutant remains a challenging task. As an alternative approach, we studied GRA18 functionality in the RAW cell line because they are transfectable cells and responded to *T. gondii* infection similarly to BMDMs with regard to *Ccl17* expression, and *Ccl22* to some extent. We agree, it is confusing that the chemokine induction differs from each other with regard to β-catenin in RAWs. However, it seems important to us that the apparent inconsistencies for data obtained in different settings and contexts could be presented as they might provide relevant information for a better understanding of the GRA18-β-catenin and chemokine functional interactions. We have tried to address this issue in the following comment:

Visibly, there are some GRA18 and β-catenin-independent effects but would argue that the major induction effect upon *T. gondii* infection of the primary BMDM is GRA18 dependent. By ectopic expression of GRA18 in RAWs, we showed that the up-regulation of both *Ccl17* and *Ccl22* was dependent on β-catenin, though with a modest contribution of β-catenin for *Ccl22* (Figure 6E). Different results were obtained upon infection: in this assay, *Ccl22* expression is poorly dependent on *GRA18*, and as expected, independent of β-catenin (Figure 6F). Conversely, parasites overexpressing GRA18 allowed a further induction of *Ccl22* (from 30 to 500 RQ) (Figure 6F). Importantly, this effect is partially dependent on β-catenin, consistent with our general conclusion about GRA18 activity relying on β-catenin, at least partially. We think that these discrepancies are reflecting a different mode of regulation of *Ccl22* between the transformed RAWs and the primary BMDMs. We currently do not have explanation for why GRA18 at physiological levels (provided by wt parasites) is not active in RAWs. We propose that the induction of *Ccl22* in the RAWs is more likely dependent on a different regulator delivered by *T. gondii*, which explains why the induction is β-catenin independent. We have modified the text (line 284-293) as well as the model proposed in Figure 8 accordingly, expectantly clarifying that the mechanism of regulation of *Ccl22* is likely less dependent on β-catenin than *Ccl17* and might differ according to the transformed (RAW) or primary (BMDM) status of the cells under analysis.

4) Subsection “Quantitative real-time PCR”: Was the RNA quality/integrity controlled and Tbp as the only control gene tested for its expression stability? Not doing so could potentially undermine the robustness and reliability of the qPCR data. Why is only 1 biological replicate shown in the qPCR data and not a combination of 2 or more?

Total RNA quality and integrity were assessed by agarose gel electrophoresis while quantification of 28S and 18S ribosomal RNA considered as the gold standard for evaluation of RNA integrity (Sambrook and Russel, 2001). Additionally, *T. gondii* infection rates were verified as judged by the parasite rRNA band that migrates in-between the 28S and 18S RNAs of the host cell. Our lab uses different genes as internal controls for qRT-PCR analysis (i.e. *Tbp, B2m, 18S* rRNA…). It is difficult to predict exactly how any given gene behaves under a given range of conditions, but the RNA-Seq data from our lab and others have shown that the *Tbp* transcripts remained constant in BMDM regardless of *T. gondii* infection and *T. gondii* strains (Melo et al., 2013).Furthermore, *Tbp* transcript levels are close to those of *Ccl17, Ccl22,* and *Ccl24* studied in the present manuscript (i.e. with RPKM ranging from 10 to 100), therefore making *Tbp* an appropriate control gene for assaying gene expression in murine macrophages. When performing the qRT-PCR, a particular attention was made to having constant Ct values across the different samples, thus confirming both RNA integrity and quantification. We certainly appreciate this concern but feel confident in the robustness and reliability of our qPCR data that confirm the RNA-Seq results. The method section now includes RNA quality and integrity assessment. To ensure that the effects of GRA18 on host cell gene expression was not dependent on the quality of the BMDMs prepared, the biological replicates were performed using different lots of BMDMs using somewhat different MOI. As expected, whilst the patterns were generally consistent, the Ct values were sufficiently different to introduce a large error when combined. It sounds more cumbersome than helpful to combine the data. So, we strongly feel that showing representative data set in these conditions is the more reliable for accurately interpreting the general trends.